# An integrated technology for quantitative wide mutational scanning of human antibody Fab libraries

Brian M. Petersen[1,3], Monica B. Kirby[1,3], Karson M. Chrispens[1], Olivia M. Irvin[1], Isabell K. Strawn[1], Cyrus M. Haas [1], Alexis M. Walker[1], Zachary T. Baumer[1], Sophia A. Ulmer[1], Edgardo Ayala[2], Emily R. Rhodes[1], Jenna J. Guthmiller [2], Paul J. Steiner[1] & Timothy A. Whitehead [1] ✉

Antibodies are engineerable quantities in medicine. Learning antibody molecular recognition would enable the in silico design of high affinity binders against nearly any proteinaceous surface. Yet, publicly available experiment antibody sequence-binding datasets may not contain the mutagenic, antigenic, or antibody sequence diversity necessary for deep learning approaches to capture molecular recognition. In part, this is because limited experimental platforms exist for assessing quantitative and simultaneous sequence-function relationships for multiple antibodies. Here we present MAGMA-seq, an integrated technology that combines multiple antigens and multiple antibodies and determines quantitative biophysical parameters using deep sequencing. We demonstrate MAGMA-seq on two pooled libraries comprising mutants of nine different human antibodies spanning light chain gene usage, CDR H3 length, and antigenic targets. We demonstrate the comprehensive mapping of potential antibody development pathways, sequence-binding relationships for multiple antibodies simultaneously, and identification of paratope sequence determinants for binding recognition for broadly neutralizing antibodies (bnAbs). MAGMA-seq enables rapid and scalable antibody engineering of multiple lead candidates because it can measure binding for mutants of many given parental antibodies in a single experiment.

The success of AlphaFold2[1] for predicting structure from sequence has spurred intense interest in deep learning approaches for protein functional prediction. Arguably the largest open prize in protein biotechnology is learning antibody molecular recognition, as this would enable the in silico design of developable, high affinity binders against any antigenic surface. Deep learning has been utilized to advance antibody design approaches for overall structure prediction[2,3], paratope and epitope identification[4], affinity maturation[5,6] and antibody sequence humanization[7]. These examples highlight the promise of

deep learning approaches but also their limitations. Put simply, unbiased experimental antibody binding datasets do not exist at the scale required for extant deep learning algorithms to capture antibody molecular recognition[8,9].

Researchers recently assessed the scale of experimental data required for accurate prediction of antibody binding effects upon mutation[9]. Through simulated data, they found that a training dataset comprising hundreds of thousands of unbiased antibody-antigen binding measurements across thousands of diverse antibody-antigen

[1]Department of Chemical and Biological Engineering, University of Colorado Boulder, Boulder, CO, USA. [2]Department of Immunology and Microbiology, University of Colorado Anschutz Medical Campus, Aurora, CO, USA. [3]These authors contributed equally: Brian M. Petersen, Monica B. Kirby. ✉e-mail: timothy.whitehead@colorado.edu

complexes would be sufficient to learn the effect of mutation on binding energetics. The structure of this data—on the order of a few hundred mutational data points per antibody spread across thousands of antibodies targeting diverse antigenic surfaces—suggests a different paradigm than deep mutational scanning approaches[10], which assess tens of thousands of mutations for individual proteins. Requirements for this 'wide mutational scanning' paradigm include the ability to (i) determine quantitative monovalent binding energetics, with measurement uncertainty, for multiple antibodies against different antigens and over a wide dynamic range, (ii) recapitulate the native pairing of variable heavy and light chains which can be achieved using antigen binding fragments (Fabs), (iii) track multiple mutations per antibody on either or both chains simultaneously, and (iv) include internal controls for quality control and validation. This technology could also be deployed immediately for current antibody engineering applications, including the reconstruction of multiple probable antibody development pathways[11], rapid affinity maturation campaigns for multiple leads simultaneously, fine specificity profiling for antibody paratopes, and antibody repertoire profiling against different immunogens.

Current antibody engineering techniques exist but have not demonstrated the ability to generate the depth of data required for learning antibody molecular recognition. Antibody deep mutational scanning using various display techniques has been demonstrated for different task-specific applications but does not provide quantitative binding information. Deep mutational scanning has been used to determine quantitative changes in binding affinity for protein binders but only for a narrow dynamic range[12,13]. TiteSeq[14] utilizes yeast surface display and next generation sequencing to ascertain quantitative affinities, but has only been demonstrated for a library from one parental antibody single chain variable fragment (scFv)[15], which can alter the paratope through the constrained folding of heavy and light chains imposed by an inserted linker[16]. Another high-throughput technique demonstrated for one antibody included high-throughput mammalian

display[17]. Additional demonstrations[18,19] exist that have evaluated multiple antibodies and antigens simultaneously but are not high-throughput.

We introduce **MAGMA-seq**, a technology that combines **m**ultiple **a**nti**g**ens and **m**ultiple **a**ntibodies and determines quantitative biophysical parameters using deep **seq**uencing to enable wide mutational scanning of antibody Fab libraries. We demonstrate the ability of MAGMA-seq to quantitatively measure binding affinities, with associated confidence intervals, for multiple antibody libraries. We validate the results of MAGMA-seq with isogenic antibody variant titrations (i.e. labeling isogenic yeast displaying Fabs at various concentrations of antigen and fitting fluorescence measurements to a binding isotherm to extract $K_D$). We further demonstrate the utility of MAGMA-seq on a mixed pool of antibody libraries with two distinct antigens, SARS-CoV-2 spike (S1) and influenza hemagglutinin (HA), and recover the sequence-binding profiles for six antibodies across four distinct protein surfaces. MAGMA-seq facilitates the engineering of antibodies for different applications in parallel: we demonstrate the mapping of potential antibody development pathways, antibody responses to multiple epitopes simultaneously, and identification of paratope sequence determinants for binding recognition for broadly neutralizing antibodies (bnAbs). MAGMA-seq enables rapid and scalable antibody engineering.

## Results

The protocol for MAGMA-seq (Fig. 1a) starts by generating mutagenic libraries for all antibodies of interest in a Fab format. Fab libraries are subcloned into yeast display vectors each containing a 20 nt molecular barcode; the Fab variant and barcode are paired by sequencing. The library is transformed into yeast, and yeast is grown and induced to surface display the Fabs. The yeast library is sorted at multiple labeling concentrations of antigen(s) by collecting a fixed percentage of yeast cells. After sorting, the collected yeast plasmids are extracted, and the barcode region is sequenced using short-read sequencing. The

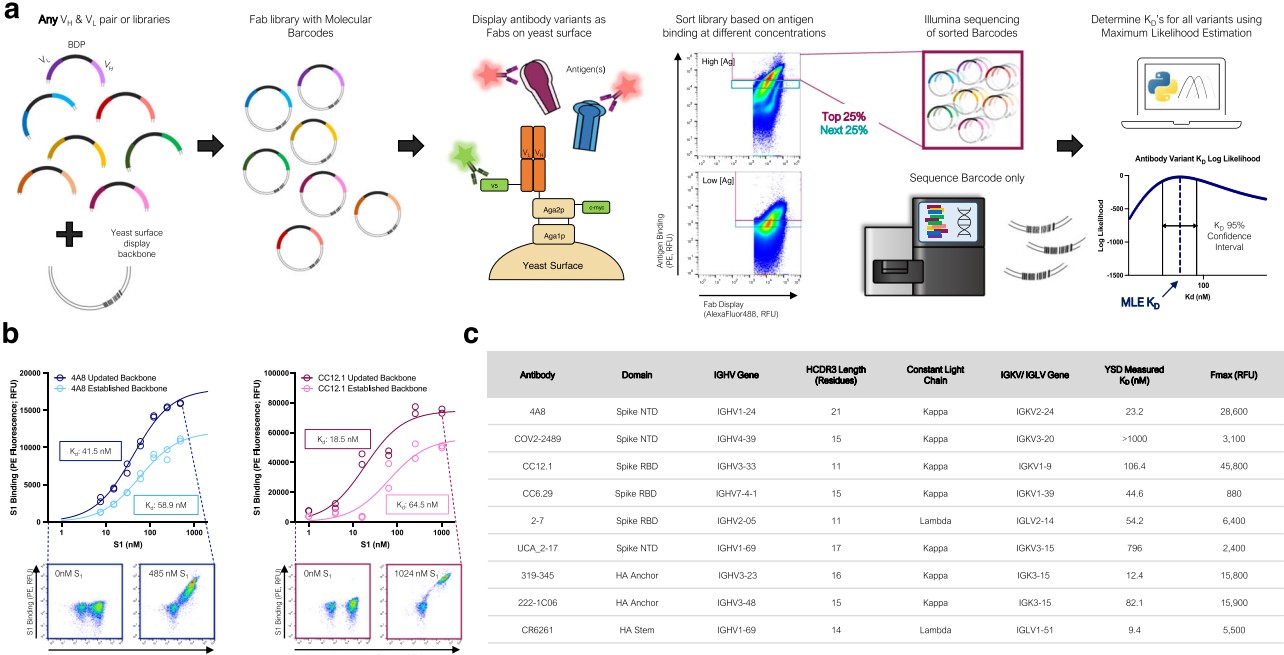

**Fig. 1 | MAGMA-seq is an integrated technology for antibody wide mutational scanning. a** Protocol schematic. **b** Yeast surface titrations of 4A8 and CC12.1 Fabs against Fc-conjugated $S_1$ in the established (light) and updated (dark) yeast surface display vectors. Cytograms from indicated data points are shown for updated yeast surface backbones. Inset describes experimentally determined

$K_D$ values ($n = 2$). **c** Antibodies assessed using updated yeast surface display vectors. RBD Receptor Binding Domain Wuhan Hu-1, NTD N Terminal Domain Wuhan Hu-1, NA influenza neuraminidase N2 A/Brisbane/10/2007, HA influenza hemagglutinin A/Brisbane/02/2018 H1. Source data are provided as a Source Data file.

sequenced data and sorting parameters are then input into a computational maximum likelihood estimation (MLE) pipeline to infer most likely biophysical parameters, and associated confidence intervals, for each antibody variant.

There have been several yeast display Fab plasmids described[20–26]; ours most closely relates to a Golden Gate compatible plasmid from Rosowski et al.[20] Common to many plasmids, including Rosowski et al.[20], is the light chain and heavy chain ($V_H$ and CH1) expressed using a Gal1/Gal10 galactose-inducible bidirectional promoter (BDP). We use Golden Gate[27] to assemble small shuttle vectors containing the $V_H$, $V_L$, and BDP, as well as regions of homology to the CH1 and light chain sequence. After mutagenesis, the Fab yeast surface display library is generated by Gibson assembly[28] using the regions of homology on the shuttle vector and empty yeast surface display vector containing the barcode. Beyond these innovations, we made several useful changes to the Rosowski plasmid (Supplementary Fig. 1), including (i) constructing plasmids for both kappa and lambda light chains; (ii) encoding a V5 C-terminal epitope tag on the light chain to assess light chain expression; and (iii) making a conservative coding mutant in CH1 and several silent mutations on the yeast vector for compatibility with short-read sequencing.

To test whether our updated plasmids interfered with Fab binding, we performed yeast surface titrations of SARS-CoV-2 antibodies 4A8[29] and CC12.1[30] against Wuhan Hu-1 S1 in the established and updated yeast surface display vectors (Fig. 1b) and fit the mean fluorescence data ($\overline{F}$) to a saturable binding isotherm:

$$\overline{F} = (F_{max} - F_{min})\frac{[S1]_o}{K_D + [S1]_o} + F_{min} \qquad (1)$$

Here $F_{max}$ is the maximum average cell fluorescence at binding saturation, $[S1]_o$ is the ligand concentration, $F_{min}$ is the cell autofluorescence, and $K_D$ is the monovalent binding dissociation constant. The confidence intervals for $K_D$ overlapped for both antibodies (Fig. 1b), suggesting that the combined changes were not deleterious for binding. For further validation, we performed additional yeast surface titrations with a representative set of antibodies encompassing diverse Complementarity-determining region (CDR) H3 lengths (lengths 11–23), immunoglobulin heavy chain variable region (IGHV) gene families, and either lambda or kappa light chains (Fig. 1c; Supplementary Fig. 2). In all cases, interpretable binding isotherms were observed. Thus, our yeast display plasmids can measure binding for a range of human Fabs.

To demonstrate the capability of MAGMA-seq to track potential development trajectories of multiple antibodies simultaneously, we selected three anti-S1 antibodies[29–31] that target Wuhan Hu-1 S1 at two distinct domains, the RBD and the NTD (Fig. 2a). For each of these antibodies, mutagenic libraries theoretically comprising all possible sets of mutations between the mature and inferred universal common ancestor (UCA) were constructed using combinatorial nicking mutagenesis[32,33] and the libraries were pooled in approximately equimolar ratios and assembled into the yeast surface vector with a target of multiple barcodes per antibody variant (Fig. 2a).

Several deep mutational scanning protocols pair a barcode to an encoded protein variant using long-read sequencing[10,34–37]. MAGMA-seq is compatible with both long-read sequencing and short-read sequencing. For short-read sequencing, the barcode is separately paired with the $V_H$ and $V_L$ using independent Golden Gate intramolecular ligation reactions[38], which places the barcode adjacent to either the CDR H3 or the CDR L3 (Fig. 2b). The reaction products are separated on an agarose gel to remove concatemers and isolate the correct intramolecular ligation product (Supplementary Fig. 3), and amplicons are prepared for paired end short-read sequencing. PCR-based amplicon preparation of mixed populations is known to result in chimera formation between closely related nucleic acid sequences[35,39]. We

evaluated several different amplicon preparation protocols by assessing chimera formation between three isogenic plasmids containing distinct mutations and unique barcodes. Using this approach, we identified a protocol resulting in low amounts of overall chimera formation (Supplementary Fig. 4).

To evaluate the fidelity of our protocol, we sequenced 20 isogenic clones using Oxford Nanopore sequencing. The pooled, mutagenic antibody library was prepared in replicates for Illumina short-read sequencing following our optimized protocol for both $V_H$ and $V_L$ pairings. 95% (19/20; replicate 1) and 85% (17/20; replicate 2) of barcode-antibody pairing was identical between nanopore and short-read sequencing (Fig. 2c), and no incorrect calls were made in either replicate. In total, we paired 1059 barcodes and recovered 64/64 CC12.1 variants (100% library coverage), 48/64 COV2-2489 variants (75% library coverage) after an alternative filtering step (Supplementary Fig. 5), and 56/64 4A8 variants (87.5% library coverage) with a mean of 4.8 barcodes per variant (Fig. 2d).

The library was transformed into yeast, passaged, and induced by galactose. We sorted the library at 10 different S1 labeling concentrations by sorting yeast cells into two bins by fluorescence using the channel corresponding to binding S1 (Figs. 1a and 2e, Supplementary Fig. 6). We sequenced and counted the number of barcodes collected from each of the bins at every sampled concentration as well as a reference population of Fab displaying cells. The count data were aggregated with fluorescence bin limits, sorted cell counts, and predetermined parameters describing the expected fluorescence distributions, and then analyzed by a custom MLE algorithm to generate monovalent binding dissociation constants ($K_D$) and max mean fluorescence at saturation ($F_{max}$) estimates for each variant. Our MLE algorithm performs minimization of the difference between observed and expected sequencing counts given an underlying system of equations describing the theoretical distributions and anticipated measurement error (for full details, see Supplementary Note 1, Supplementary Figs. 11-13, Supplementary Table 1). Importantly, the algorithm can quantify $K_D$ estimate uncertainty (Fig. 2f). Distributions of $K_D$ estimates were observed to be consistent across barcodes of the same variant, with high overlap between confidence intervals (Fig. 2g and Supplementary Fig. 7). Our MLE algorithm uses two fixed global parameters relating to the estimated error rate in FACS and the fluorescence probability distribution of the expressed constructs. We evaluated the sensitivity of the output on these parameters, finding that the mean absolute error in log$K_D$ ratio ranged from 0.016 - 0.039 $\log_{10}(K_D/K_{D,wt})$, showing little effect overall on our parameter choices (Supplementary Fig. 8).

To address whether parameter estimates from MLE are consistent with isogenic titrations, we used combinatorial nicking mutagenesis[32] to prepare biological replicates for 61 separate 4A8 variants. For each variant, we performed four isogenic titrations ($n = 4$; 2 technical replicates and 2 biological replicates of each, see Supplementary Data 2) and determined the change of dissociation constant relative to the mature 4A8 Fab ($\log_{10}(K_D/K_{D,wt})$). While we observed a single outlier, likely because of low sequencing coverage (average counts per bin = 7, Fig. 2h), the mean absolute error of MLE generated $K_D$s relative to the isogenic titrations fell at or below the level of precision of the isogenic titrations for almost all variants tested (isogenic titration experimental limit = 0.21 log$K_D$/log$K_{D,ref}$, Fig. 2h). Additionally, the MLE algorithm captured the statistically significant differences in $F_{max}$ that are known to exist between 4A8 and CC12.1 Fabs from isogenic titrations (Fig. 2i). Thus, MAGMA-seq can recover biophysically meaningful parameters that are consistent with isogenic titrations.

We performed regression analyses on the MAGMA-seq output to gain insight into the impact of individual mutations as well as to determine epistatic effects of mutations on the overall development trajectory for the 4A8, CC12.1, and COV2-2489 antibodies. As expected, due to the high $K_D$ and low $F_{max}$ observed for COV2-2489 WT (see

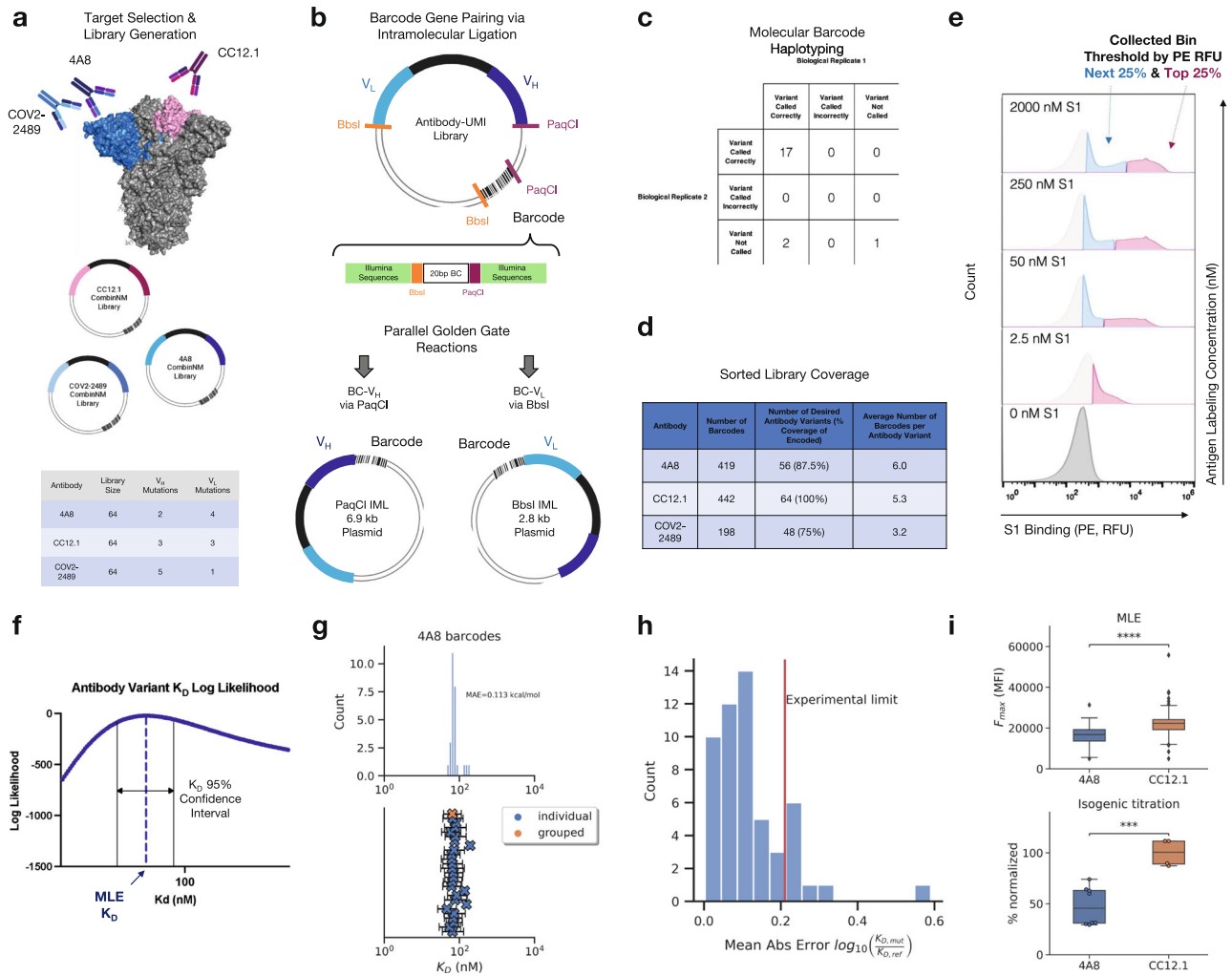

**Fig. 2 | Validation of barcode pairing and parameter estimation for MAGMA-seq. a** Mutagenic library contains 192 variants of 4A8, COV2-2489 (NTD targeting), and CC12.1 (RBD targeting) Fabs. **b** Molecular barcode in yeast display plasmid backbone allows for barcode pairing by intramolecular ligation followed by short-read sequencing. **c** Barcode pairing method achieves correct variant calls confirmed by ONT sequencing. **d** Barcode and variant coverage of haplotyped libraries. **e** Examples of gating thresholds for FACS sorting of library for 4/10 of the sampled antigen concentrations. Top 25% bin shown in pink and next 25% bin shown in blue. **f** MLE quantifies $K_D$ uncertainty via confidence interval calculation. **g** MLE $K_D$ estimates for all barcodes haplotyped as 4A8 WT (top) with 95% confidence intervals

for each barcode (blue X, n = 26) and grouped barcodes (orange X) (bottom). **h** Mean absolute error for MLE $K_D$ estimates for counts collapsed by variant versus isogenic titration values (4A8 only). **i** Maximum mean fluorescence values ($F_{max}$) for 4A8 and CC12.1 antibodies calculated via MLE in absolute terms (top; 4A8: n = 70, CC12.1: n = 83) and isogenic titration as a percentage normalized by the CC12.1 average (bottom; 4A8: n = 8, CC12.1: n = 4). Quartiles are shown for all data except outliers defined as outside 1.5*IQR, p-values calculated by two-sided Welch's t-test (***: 1e-4 < p <= 1e-3, ****: p <= 1e-4, MLE: p = 1.7E-8, Isogenic: p = 4.9E-4). Source data are provided as a Source Data file.

Fig. 1c), we noticed that few barcodes from any variants of this antibody appeared in any of the sorted bins at substantial quantities and similar analysis was not completed. For 4A8 and CC12.1, we performed one-hot encoding of the programmed mutations and then analyzed each antibody separately using different regression techniques (Ordinary Least Squares (OLS), Least Absolute Shrinkage and Selection Operator (LASSO)[40], and Ridge Regression[41] (Supplementary Data 3). While agreement was observed amongst all regression methods, we selected the LASSO due to the parameter minimization inherent to the method.

LASSO regression for the 4A8 isogenic clone $\log K_D$ ratio titration data and a $2^{nd}$ order model fit the data with MAE = 0.099 $\log_{10}(K_D/K_{D,wt})$. All $2^{nd}$ order coefficient weights fell below 0.07 $\log_{10}(K_D/K_{D,wt})$ (less than 17% absolute difference in binding affinities), supporting a sparse development pathway (Fig. 3a). An identical analysis performed on the 4A8 MLE dataset reproduced the same sparse pathway results (Fig. 3a; MAE = 0.063 $\log_{10}(K_D/K_{D,wt})$). Surprisingly, only the light chain mutation M94T had any appreciable effect on binding. The coefficient

weights for the 4A8 titrations and MLE proved consistent with a correlation coefficient of 0.94 for all first order weights (Fig. 3b). The correlation coefficient for all first and second order weights is lower at 0.70 due to the noise present in the titration data collection (Fig. 3b). MAGMA-seq also allowed us to perform regression analysis on $F_{max}$, a proxy for the total amount of active Fab on the yeast surface. For 4A8, a $2^{nd}$ order model showed $F_{max}$ is influenced by multiple mutations. I59M decreases $F_{max}$ by 10%, and K120Q improves $F_{max}$ values by 9% compared to mature 4A8 (Fig. 3c).

Analogous regression for antibody CC12.1 was performed using the MLE data for $\Delta\Delta G_{binding}$ and $F_{max}$. A second order model described the data with MAE = 0.07 $\log_{10}(K_D/K_{D,wt})$ and 3923 RFU for $\Delta\Delta G_{binding}$ and $F_{max}$, respectively. Consistent with 4A8, we found a sparse mutational landscape with CC12.1 and S1 where only two mutations, F27L and Y58F, are required for enhanced affinity (Fig. 3d). M104L improves $F_{max}$ values by approx. 16% in the presence of F27L and Y58F (Fig. 3e).

4A8 binding to S1 is mediated predominantly by the $V_H$ chain with important contacts to the NTD in CDR H1 and CDR H3[29]. $V_L$ M94T is the

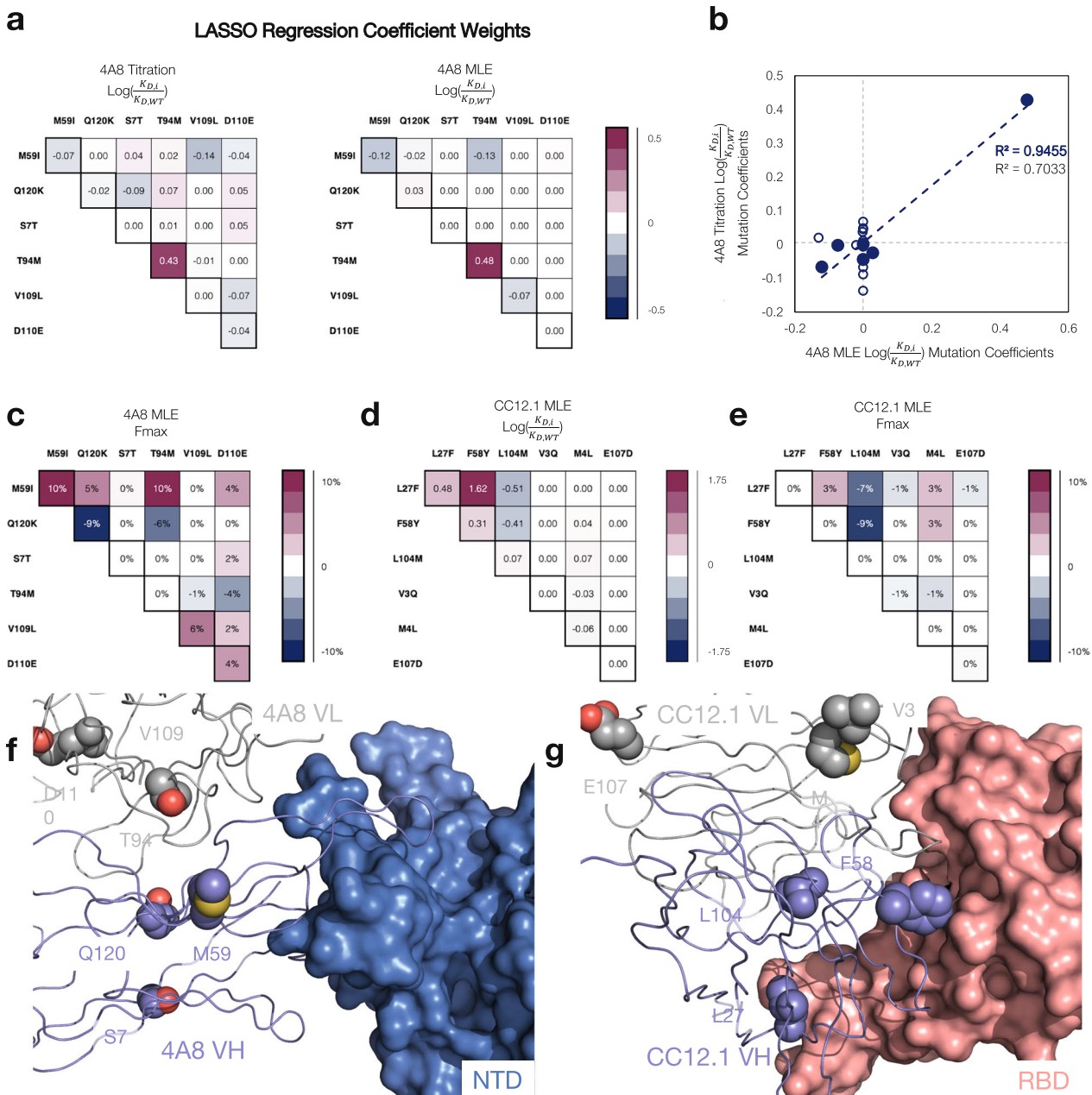

**Fig. 3 | Antibody development landscapes for 4A8 and CC12.1 are sparse.**
**a** Comparison of 4A8 one body and two body parameter binding affinity weights inferred from (left) isogenic titrations and (right) MAGMA-seq. Binding affinities are represented as $\log K_D$ ratios relative to the mature antibody sequence. **b** Correlation between isogenic titrations and MAGMA-seq parameter weights. Blue closed circles are one-body weights, and open circles are two-body weights. **c** Parameter weights

for 4A8 $F_{max}$ percentage differences relative to the mature antibody. **d**, **e** CC12.1 MLE parameter weights for **d** $\log K_D$ ratios and **e** $F_{max}$ as inferred from MAGMA-seq. **f**, **g** Structural complexes of SARS-CoV-2 Wuhan Hu-1 S antibodies **f** 4A8 bound to NTD (PDB ID: 7C2L), and **g** CC12.1 bound to RBD (PDB ID: 6XC3). Positions mutated from the inferred UCA sequence are shown as purple ($V_H$) or gray ($V_L$) spheres. Source data are provided as a Source Data file.

only one of six mutations from germline that improves binding affinity. A structural hypothesis for this mutation is that it repositions the CDR H3 in a more productive conformation for NTD recognition (Fig. 3f). CC12.1 uses both $V_H$ and $V_L$ to contact S RBD[42] (Fig. 3d). Y58F directly contacts the RBD surface for improved binding, while F27L may subtly reposition the CDR H1 for improved recognition. M104L decreases binding affinity in the context of F27L and Y58F but improves functional expression, and it may participate in subtle antibody-antigen rearrangements which could cause the minor 2-body effects seen (Fig. 3d, g). MAGMA-seq alone as well as in combination with known structures can aid in the structural and genetic understanding of antibody development trajectories.

To determine whether MAGMA-seq can evaluate multiple antibodies sorted against multiple antigens simultaneously, we prepared a library containing mutants of eight distinct antibodies[29,43–46] (1G01, 1G04, 319-345, 222-1C06, CR6261, 2-7, UCA_2-17, and 4A8) containing varying light chain gene usage and CDR H3 length. 1G01 and 1G04 bind at the active site on NA influenza neuraminidase N2 A/Brisbane/10/2007[43]. CR6261 is a bnAb binding to group I HAs[45]. 319-345 and 222-1C06 are nAbs which recognize the anchor epitope on H1 HA[44]. 2-7, UCA_2-17, and 4A8 recognize SARS-CoV-2 spike Wuhan Hu-1[29,46] (Figs. 1c, 4a). We sorted replicates of this library of 4,105 matched barcoded antibodies against 11 varying combined concentrations of HA and S1. The 11 sorts were structured such that, at all labeling

concentrations, the average population had an appreciable binding signal (Supplementary Fig. 9). One labeling concentration contained only HA or S1, respectively. Additionally, the library contained internal controls for evaluating the sorting error and for assessing the fidelity of affinity reconstruction. The complete dataset for all antibody variants is listed in Supplementary Data 4. As expected, none of the NA-specific 1G01 or 1G04 antibody variants had inferred dissociation

constants below 1 μM for either the HA or S1 antigen. HA-specific and S1-specific antibodies mapped neatly to one of the two antigens using the antigen-only sort (Fig. 4b). The 4A8 variants, included as internal controls, were consistent with the parameter weights from the previous sort (logK$_D$ ratio of T94M relative to the S7T variant: 1.33). Additionally, the estimated K$_D$ values from MLE are reasonably consistent between replicate sorts. After removing variants containing

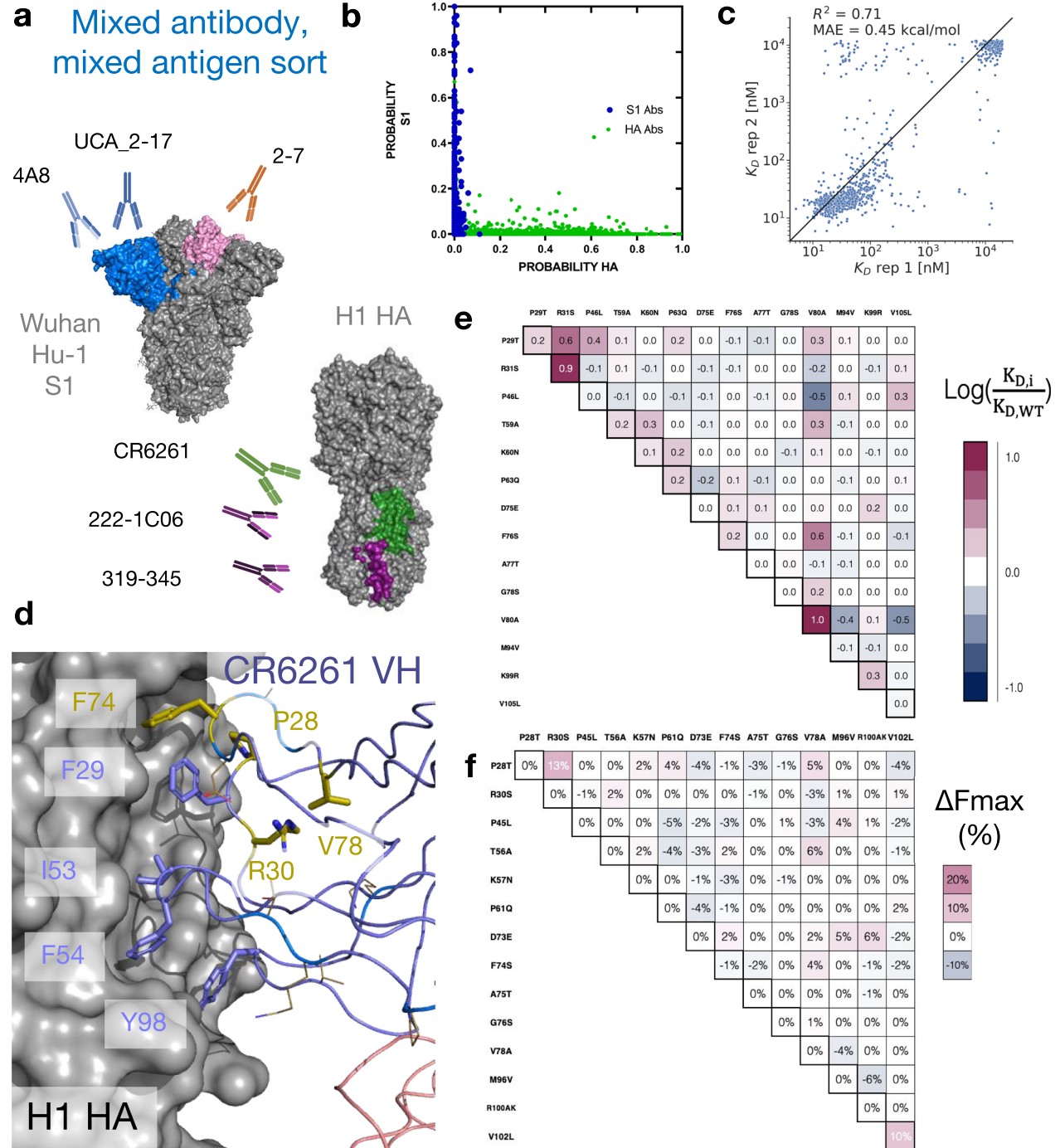

**Fig. 4 | MAGMA-seq infers biophysical properties in mixed antibody, mixed antigen sorts. a** Antigen-specific antibody sub-libraries and antigens used in the sorting experiments. Non-binders 1G01 and 1G04 were also included. **b** Probability of an antibody variant sorted into an antigen-specific bin when only 2000 nM of S1 (Y-axis) or H1 HA (X-axis) was incubated with yeast. For HA, only the top sorting bin was included in the analysis. **c** R$^2$ of MLE K$_D$ estimates from sort replicates for HA targeting antibodies 222-1C06, 319–345 (anchor epitope), and CR6261 (stem

epitope). **d** Structure (PDB ID: 3GBN) of CR6261 bound to H1 HA. Purple sticks are HA-contacting positions that are encoded from the inferred UCA sequence. The side chains of residues mutated in the mature antibody relative to the UCA are shown as gold sticks and lines. **e, f** LASSO regression of **e** logK$_D$ ratios and **f** F$_{max}$ percentage differences one body and two body weights for CR6261. Weights are shown relative to the mature antibody. Source data are provided as a Source Data file.

stop codons and non-converged values, we observe an $R^2$ of 0.71 and MAE of 0.32 $\log_{10}(K_D/K_{D,wt})$ for antibodies 222-1C06, 319–345 (anchor epitope), and CR6261 (stem epitope) (Fig. 4c). Relative to replicate 1, replicate 2 underpredicts some of the intermediate affinity antibodies. We attribute this discrepancy to the absence of the 100 nM labeling bin for the second replicate.

Two antibodies in the library contained mutations allowing for the reconstruction of potential development trajectories from their inferred UCA sequence. 2–7 is a Wuhan Hu-1 S1-specific nAb[46]. 2–7 contains five mutations from its inferred germline, all in the $V_L$ (A5G, A31G, D52E, K55N, T95S). The inferred development (Supplementary Fig. 10) was superficially like the sparse development pathways observed for 4A8 and CC12.1, with four 1st order couplings predicting the dissociation constants of the potential pathway variants, as supported from LASSO regression.

CR6261 is an influenza bnAb targeting the HA stem epitope originally described by Throsby et al.[45]. It is an unusual antibody in two ways. First, its development trajectory is dissimilar to other VH1-69 anti-HA antibodies previously characterized[47]. Second, it confers molecular recognition only through its $V_H$, mainly by positioning apolar residues at framework 3 (FR3) (F74, V78), in CDR H1 (F29), and in CDR H2 (F54) in a hydrophobic groove[48,49] (Fig. 4d). Both CDR residues are encoded in germline VH1-69 sequence in allelic human populations, but the inferred UCA sequence does not appreciably recognize the H1 HA stem epitope[50]. The potential first steps of its trajectory from its inferred UCA sequence have been developed by Lingwood et al.[50], supporting a first committed step of some combination of H1 mutations T28P and S30R necessary for orientation of F29, and at least some subset of the framework 3 (FR3) mutations (E73D/T75A/S76G/A78V) necessary for F74 insertion (Fig. 4d). We sampled 2.9% (470/16,384 possible variants) of the potential sequences between the UCA and mature CR6261. MAGMA-seq recovered a $K_D$ of 12 nM for mature CR6261, consistent with isogenic titration of 9.4 nM and with previous literature reports[45]. LASSO regression supported a 2nd order epistatic model (Supplementary Data 3), with a total of 5 1st order and 15 2nd order weights above an absolute 0.18 $\log_{10}(K_D/K_{D,wt})$ energetic threshold. Consistent with the studies from Lingwood and Pappas, the strongest 1st order weights contributing to binding affinity are T28P, S30R, and FR3 mutation A80V (Fig. 4e), and the two strongest 2nd order weights are the epistatic couplings between T28P/S30R (0.65 $\log_{10}(K_D/K_{D,wt})$) and S74F/A80V (0.57 $\log_{10}(K_D/K_{D,wt})$). The known epistasis in the T28P/S30R mutations can be rationalized as altering the orientation of the CDR H1 loop such that F29, usually buried, largely becomes solvent exposed in the unbound structure. Consistent with this hypothesis, the surface expression of Fabs containing T28P/S30R mutations decreased by approximately 10% (Fig. 4f), as expected for mutations which increase the apolar solvent accessible surface area. The other epistatic relationship observed of S74F/A78V can relate to the positioning of hydrophobic residues, where the 78V is needed to constrain the correct F74 rotamer for precise shape complementarity in the stem groove. In sum, the sparse sampling of bnAb mutants allow for the reconstruction of the development pathways that are in concordance with the existing body of structural, genetic, and immunological evidence for this antibody. Thus, MAGMA-seq can reconstruct the likely development pathways for multiple human antibodies against different antigens in the same experiment.

The libraries described thus far are all retrospective analyses of antibody development trajectories, where libraries encoded chimeras of the mature and UCA sequences. To further investigate the utility of this method, our second demonstration of MAGMA-seq included a prospective antibody development library and a few CDR targeted site-saturation mutagenesis libraries. We generated each of these antibody libraries in parallel reactions and subsequently pooled and barcoded the variants. We bottlenecked the library, which selected individual variants randomly, and assessed it with MAGMA-seq.

To test whether MAGMA-seq could map prospective antibody development trajectories, the mixed library contained a subset of a larger library of the UCA sequence of the anti-S NTD 2-17 (UCA_2–17)[46]. This larger library theoretically contained all single nucleotide substitutions at the CDRs and framework positions. Its UCA was predicted to bind at a $K_d$ of 2050 nM (range 400–3200 nM; 0.23 $\log_{10}(K_D/K_{D,wt})$ s.d.; 58 barcoded UCA sequences) and a mean $F_{max} = 890$, consistent with measurements of the isogenic control (Fig. 1c). We were able to recover 318 uniquely barcoded variants. Many of these mutants, like $V_H$:Y91DHN or $V_H$:C92GFY near the CDR H3, are expected to structurally destabilize the protein, resulting in non-specific binders. Still, several mutants had lower inferred dissociation constants or higher $F_{max}$ values than the UCA, including $V_H$:I51N in CDRH2 ($K_d$ 970 nM) and $V_L$:N32D ($F_{max}$ 4,000) observed in the mature 2-17 sequence, $V_H$:S75P ($K_d$ 400 nM), and $V_H$:A97P in CDRH3 ($K_d$ 490 nM) (Fig. 5a). Thus, MAGMA-seq can evaluate potential forward trajectories for antibodies that are consistent with genetic and structural data.

We also used MAGMA-seq to infer the preliminary rules of recognition for an emerging class of influenza neutralizing antibodies. Antibodies 319–345 and 222-1C06 target a distinct anchor stem epitope of H1 HA[44]. Anchor bnAbs appear to be germline restricted to light chains VK3–11 or VK3–15, with heavy chains from germlines VH3-23, VH3-30/VH3-30-3, and VH3-48. All mature anchor bnAbs encode a CDR H3 of diverse amino acid sequences, with a glycine either at the beginning or end of the CDR H3 and two to four hydrophobic residues at the middle of the sequence. The cryo-EM structure of 222-1C06 bound to H1 HA shows the structural basis of recognition. The interaction at the anchor epitope is dominated by multiple hydrophobic interactions across the heavy and light chains. The germline-encoded and invariant CDR KL3 'NWPP' motif from positions 93–95 A are at the center of the binding interface. CDRH2 (Leu55) and CDRH3 (Trp99, Pro100, Thr100a) all contribute hydrophobic contacts at the binding interface (Fig. 5b, c).

We recovered 183 and 390 single non-synonymous mutants of 222-1C06 and 319–345, respectively (1429 uniquely barcoded variants). The observed $K_D$ for mature antibodies were low nM (319–345: 16 nM; 222-1C06: 27 nM) and highly reproducible between independent barcodes (319–345: 0.092 $\log_{10}(K_D/K_{D,wt})$ s.d., $n = 171$; 222-1C06: 0.07 $\log_{10}(K_D/K_{D,wt})$ s.d., $n = 92$). CDR loops L1, L2, and H1 make peripheral contacts at the interface. Consistent with this, only 3.8% of single mutants (2/118 and 11/161 for 222-1C06 and 319–345, respectively) at CDR L1, L2, and H1 positions disrupted binding affinity by greater than 0.7 $\log_{10}(K_D/K_{D,wt})$ (Supporting Data 4). This contrasts with CDR H2, where 40% (20 of 51) of single and double mutants disrupted binding greater than 0.7 $\log_{10}(K_D/K_{D,wt})$, supporting the importance of H2 in recognition of the anchor epitope (Fig. 5c). While the library under sampled CDRH3, mutations at Trp99 for 222-1C06 (W99E $\log(K_{D,i}/K_{D,WT})$ 1.9) and Gly100d for 319–345 (G100dL/I > 2.1 $\log_{10}(K_D/K_{D,wt})$) were deleterious, consistent with the precise positioning of the loop needed for binding. In the KL3 'NWPP motif', observed mutations at N93 seem to have little effect on binding affinity, while mutations at W94, P95, and P95a seem to drastically disrupt binding in 222-1C06 (Fig. 5c). Intriguingly, mutations at these same positions in 319–345 are only mildly deleterious (Fig. 5c), suggesting that the antibody paratopes are positioned slightly differently against HA.

To identify candidate mutants with lower binding affinities than the mature antibodies, we identified all variants with $\log(K_{D,i}/K_{D,wt})$ values falling at least two standard deviations below zero. No 319-345 mutants met this cutoff, while four 222-1C06 variants did (VH:E100bG, VH:S54G, VH:D101G, and VH:D101S; Fig. 5c). E100b is adjacent to an acidic patch on HA in the structural complex (Fig. 5c), and so mutation to glycine likely improves binding by eliminating this unfavorable electrostatic contact. The mechanistic basis of the D101 mutations remains unclear, as mutation likely disrupts a salt bridge with CDRH3 R94. Likewise, the effect of S54G is obscure, although we

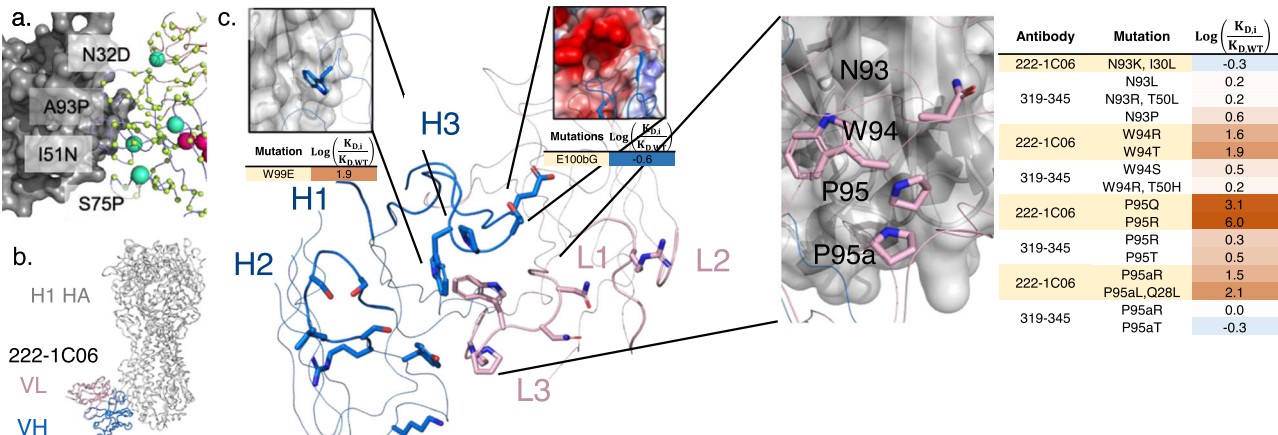

**Fig. 5 | MAGMA-seq samples the function sequence-binding landscape for neutralizing antibodies. a** Forward trajectories of the UCA of anti-S1 nAb 2-17. The sampled library is a subset of all potential single nucleotide substitutions in both VH and VL. All sampled positions are shown with CA atoms shown as lime spheres. Larger cyan spheres encode gain of function antibody variants. **b** Previously solved structure of 222-1C06 bound to H1 HA (PDB ID 7T3D). **c** 222-1C06 paratope and mutational profiles for certain residues in the CDR H3 and KL3. CDRs L1-L3 and H1-H3 are shown as larger width ribbons than the rest of the main chain. Residues with a CB within 5 Å of HA are shown as colored sticks. The panel inset for the E100bG mutation shows the electrostatic potential surface of H1 HA. Source data are provided as a Source Data file.

note that this mutation occurs in several 319–345 clonotypes[44] isolated from patients.

## Discussion

In this paper we present MAGMA-seq, an integrated technology for quantitative wide mutational scanning of human antibody Fab libraries. We demonstrate MAGMA-seq on two pooled libraries comprising mutants of ten different human antibodies spanning light chain gene usage, CDR H3 length, and antigenic targets. Analysis of MAGMA-seq outputs allows for the simultaneous mapping of retrospective and prospective potential antibody development pathways, paratope affinity maturation, and the sequence dependence on binding for broadly neutralizing antibodies. MAGMA-seq can be deployed immediately not only in these areas but for affinity maturation campaigns, specificity mapping campaigns, and for fine paratope mapping. A compelling advantage of MAGMA-seq is its ability to measure binding for mutants of many given parental antibodies in a single experiment. Since modern biotech campaigns typically use dozens of candidates in initial testing, MAGMA-seq enables the streamlining of such measurements.

We used MAGMA-seq to reconstruct potential development pathways for anti-influenza (CR6261) and anti-SARS-CoV-2 (4A8, CC12.1, 2-7) nAbs. We found that these development pathways can be reconstructed by considering binding contributions from only a handful of the mutations. This is supported by a body of evidence from other protein families[51,52] showing the sparseness of functional protein landscapes[53]. We also found that these sequence-binding fitness landscapes were most consistent with one-body or at most two-body interactions, consistent with recent protein engineering literature[54–56]. The resulting implication is that sampling of a small percentage of potential variants is sufficient for reconstruction of fitness landscapes. Indeed, for the CR6261 experiments we sampled 470 out of 16,384 possible variants and were still able to reconstruct a development trajectory supported by existing evidence. Likely many such antibody trajectories can be inferred from relatively few experiments.

We also evaluated the sequence dependence of two recently described nAbs targeting the anchor epitope on influenza HA. Our broad findings established the importance of several key mutations at the antibody side of the interface, identified electrostatic complementarity as a mechanism for improving nAb recognition to the anchor epitope, and highlighted the importance of shape complementarity for the diverse CDR H3 sequences found to fit in the

interface. We anticipate that MAGMA-seq will be used to enumerate the sequence determinants for entire sets of antibodies targeting key neutralizing or other important antigenic epitopes.

There are some limitations with the current demonstration of this technology. First, we assess binding using yeast surface display, limiting the practical dynamic range of binding affinities to 0.5 nM–2 μM. At high affinities, the labeling time to reach equilibrium reaches >10 h, and at low affinities the antigen can dissociate off the yeast surface during sorting[57]. Many therapeutic antibodies with low picomolar monovalent binding affinities would be impracticable to assess accurately. Second, we have measured order of magnitude differences in $F_{max}$ values between different mature Fabs (see Fig. 1c). Evidence suggests some correlation between functional expression on the yeast surface and stability of variants deriving from the same parental sequence[6,58], but a complete understanding of what drives differential Fab expression *between* parental Fabs is not yet known. The low $F_{max}$ values of some antibodies can hinder MLE performance solely due to the variant having low probability of being sorted into a bin, which was exemplified by the low counts of antibody COV2-2489 variants collected in our first demonstration. Third, the MLE algorithm uses one global parameter that cannot be measured during the experiment. Despite this limiting assumption, the inferred monovalent dissociation constants match published results where known. Fourth, no explicit removal of non-specific binders, like that seen for the anti-S NTD 2-17 sorts, were performed here. A parallel sort with polyspecificity reagent could improve discrimination of bona fide binders. Fifth, we note that, due to the implementation of FACS with yeast display, accuracy of MAGMA-seq estimated binding affinities may not precisely match gold-standard in vitro measurements like SPR, where antibody/antigen interactions are more directly quantified. Additional encumbrances to the method presented include the formation of antibody sequence chimeras during intramolecular ligation that reduce the number of identified barcodes and the use of TruSeq small RNA single 6-nt index adapters that allow for more index hopping during Illumina sequencing. Technical improvements would remain compatible with the rest of the MAGMA-seq workflow. Long-read sequencing is becoming increasingly inexpensive and more accurate, and as it improves it removes the necessity of PCR amplification.

We demonstrated this technology on libraries of fewer than 10,000 variants, although the functional limit on the library size is much larger. The potential complexity bottlenecks for library size are

through generation of individual mutagenic libraries, Gibson assembly into barcoded yeast display plasmids, transformation into yeast, sorting in yeast, and sequencing. An additional complexity bottleneck arises through the linking VH and VL genotypes via barcodes. The major bottleneck at the current stage of development is through cell sorting. For sorting speeds of commercially available cell sorters, the protocol leads to approx. 1000 cells collected per sorting bin (10,000 events per second × 40% Fab displaying cells per event × 25% collection of the Fab displayed cells). Since we sample at least 150-fold above the theoretical size of the library, this means that a library size of 10,000 would take 25 minutes per labeling concentration. Sorting the full suite of 10–12 labeling concentrations would then take a full working day, including start-up and shutdown. Significantly larger libraries would require multiple days of sorting or multiple cell sorters running in parallel.

Massively parallel measurements of protein binding affinities can be used to train deep learning models to capture antibody molecular recognition. We have demonstrated that this MAGMA-seq technology can perform wide mutational scanning for multiple antibodies against different antigens over a wide dynamic range of binding affinities. These measurements are made in a natural human Fab background and have multiple internal controls needed for quality control and validation. The next steps are an integrated computational and experimental appraisal of the quality and quantity of data needed for such purposes.

## Methods

### Materials

All media components were purchased from ThermoFisher or VWR. All enzymes were purchased from New England Biolabs unless otherwise specified. The recombinant SARS-CoV-2 Spike S1-hFc-His tagged protein used for titrations and sorting was purchased from ThermoFisher (RP-876-79). The recombinant neuraminidase (NA) for titrations was obtained through BEI Resources, NIAID, NIH: N2 Neuraminidase (NA) Protein with N-Terminal Histidine Tag from Influenza Virus, A/Brisbane/10/2007 (H3N2), Recombinant from Baculovirus, NR-43784. The ectodomain of A/Brisbane/02/2018 H1 HA with a foldon trimerization domain was expressed in HEK293T cells (ATCC, female, #CRL-11268) and purified using Ni-NTA affinity chromatography. Recombinant neuraminidase and recombinant hemagglutinin were biotinylated in a 20:1 molar ratio of biotin to antigen with EZ-Link NHS-Biotin (ThermoFisher, 20217) following the manufacturer's instructions.

### Plasmids

All plasmids were constructed using either NEBuilder HiFi DNA Assembly Master Mix (New England Biolabs) for Gibson assembly[28], by Golden Gate assembly[27,59], using a Q5 Site-Directed Mutagenesis Kit (New England Biolabs), or by nicking mutagenesis[32,60]. Synthetic DNA was ordered either as gBlocks or eBlocks (IDT). A complete list of plasmids, libraries, gene blocks, and primers are located in Supplementary Data 1.

pMBK046, the old 4A8 Fab YSD vector, was constructed by Golden Gate assembly with plasmids pMBK008 and pMBK027 and gene blocks 7 & 8. pMBK047-pMBK228, the 4A8 Fab YSD library plasmids, were generated by combinatorial nicking mutagenesis[32] with pMBK046 as the template and primers 142 and 328-339 and isolated by Sanger sequencing (Genewiz) of individual colonies.

The mini-mutagenesis shuttle vectors for the 4A8/CC12.1/COV2-2489 library: pMBK231, pMBK233, pMBK234, pMBK235, pMBK236, pMBK237, pMBK248 were generated by golden gate assembly of the antibodies corresponding VH and VL gene fragments with pBMP103-UMI and pBDP and were sequence confirmed by Oxford nanopore sequencing (Plasmidsaurus).

The isogenic yeast display Fab plasmids were constructed by yeast homologous recombination. The sequence verified shuttle vector(s)

(pMBK231–pMBK248, pMBK317–pMBK318, pMBK341–pMBK342) and the yeast display plasmids pBMP103 for kappa antibodies and pMBK275 for lambda antibodies were separately digested with NotI-HF and bands corresponding to the antibody Fab or yeast vector backbone were fractionated on an agarose gel and extracted using Macherey-Nagel NucleoSpin® Gel and PCR Clean-up kit (740609.50). The purified DNA was mixed in a 2:1 molar ratio of Fab insert to yeast display backbone and co-transformed into chemically competent EBY100 (ATCC MYA-4941).

The Fab shuttle vectors with kanamycin (pBMP101 and pMBK272) and chloramphenicol (pMMP_kappa and pMMP_lambda) antibiotic resistance genes were constructed using either gBlocks or eBlocks (IDT) and Gibson Assembly using NEBuilder HiFi DNA Assembly Master Mix.

The lambda mRFP yeast surface display plasmid, pYSD_lambda_mRFP, was constructed by first digesting pYSD_kappa_mRFP with PacI and NotI-HF to remove the kappa light chain segment and next the lambda light chain e-block was cloned in by Gibson Assembly using NEBuilder HiFi DNA Assembly Master Mix.

### Construction of Fab libraries

Fabs were diversified either by complete combinatorial mutagenesis[32], site-saturation mutagenesis[61], or oligo pool nicking mutagenesis[62]. Complete combinatorial libraries of Fabs were prepared from mature human antibodies and their inferred universal common ancestor (UCA). UCA sequences were inferred using IgBLAST[63]. In total, 10 mutagenic libraries were prepared (all library details are in Supplementary Data 1).

To generate libraries L001, L002, L003, L024, and L029, combinatorial mutagenesis was performed exactly as previously described[32] using the mini-mutagenesis plasmids pMBK234, pMBK235, pMBK236, pMBK237, pMBK233, pMBK248, pMBK317, and pMBK318, as the parental plasmid DNA templates and the mutagenic oligos 137, 139, 142, 145, 146, 153, 332–339, 347–356, 375–376, and 456–459, (IDT) containing degenerate codons that encode either for the mature antibody residue or the UCA residue.

Briefly, a plasmid strand is selectively nicked and degraded leaving circular single-stranded parental template DNA strands. Mutagenic oligos containing degenerate codons can anneal to the single-stranded parental templates and the rest of the strand is synthesized and ligated via polymerase and ligase resulting in heteroduplex plasmids. Next, the alternate plasmid strand is selectively nicked and degraded and subsequently an oligonucleotide can anneal to a non-mutagenic region and the complete plasmid can be synthesized with polymerase and ligase.

To construct libraries OMIL004, OMIL006, OMIL011, and OMIL0012 for 319-345, 222-1C06, 1G01, and 1G04, targeted site-saturation mutagenesis was performed by the method of Bloom[61]. Template linear PCR products were made using primers OMI1009 and OMI1011 which amplify the VH-BDP-VL region of plasmids OMI0014, OMI0016, OMI0020, and OMI0021. 25 μL of 2× Q5 Master Mix, 2.5 μL of 10 μM OMI1009, 2.5 μL of 10 μM OMI1011, 1.2 μL of 5 ng/μL of the template plasmid were combined with 18.8 μL of water. Each of the PCR reactions were run on a thermocycler at the following settings:

1. 98 °C for 30 s.
2. 98 °C for 10 s.
3. 72 °C for 56 s.
4. Repeat steps 2 and 3 for 24 additional cycles.

PCR products were then purified over agarose gels using a NucleoSpin® Gel and PCR Clean-up kit (Macherey-Nagel, 740609.50). These products were used as templates for the forward and reverse fragmenting reactions prepared as described[61] with the following modifications: 2× Q5 Master Mix was substituted for 2× KOD Hot Start Master Mix, OMI1009 was used as the outer forward primer, OMI0011 was used as the outer reverse primer, and mutagenic oligos tiling the

VH and VL CDRs were used for the forward and reverse pools. Each fragmenting PCR reaction results in many different length products containing zero, one, or multiple codon changes. The reactions were run for 10 cycles of the above thermal cycler program. The products from this reaction were used as templates for the joining reaction. The joining reaction was prepared as described in Bloom with the same modifications from the fragmenting reactions to synthesize the full length dsDNA mutagenic genes. Additionally, the joining reaction volume was scaled from 30 µL to 50 µL to increase product yield. The joining reaction was cycled using the above program for 20 total cycles. The products from this reaction were gel purified over agarose gels using a NucleoSpin® Gel and PCR Clean-up kit (Macherey-Nagel, 740609.50).

L030, the UCA_2-17 forward trajectory library was generated following oligo pool mutagenesis[62] with a fast anneal step performed with the oligo pool listed in Supporting File 1. The molar ratio of template DNA to oligonucleotides was adjusted to 10:1. This procedure is identical described above for the combinatorial nicking mutagenesis except instead of having excess primers that anneal to each prepared single-stranded DNA parental template, the ratio of templates to primers is modified to promote an average of only one primer annealing per template to result in libraries containing variants with an average of one mutation.

A total of 1 µg of plasmid libraries in a reaction volume of 20 µL in rCutSmart buffer were separately digested with 20 units of NotI-HF (NEB) for 1 h at 37 °C. In parallel, 2–5 µg of pBMP103-UMI or pMBK275-UMI library in a reaction volume of 20 µL in rCutSmart buffer was also digested with 20 units of NotI-HF for 1 h at 37 °C. The digested DNA was fractionated on a 1 (w/v) % agarose gel. Bands corresponding to the VH-BDP-VL region of the antibodies (1.8 kB) and the yeast surface display vector backbone for the UMI library (6.4 kB) were extracted using Macherey-Nagel's NucleoSpin® Gel and PCR Clean-up kit (Macherey-Nagel, 740609.50).

The yeast surface display and barcoded mutagenic antibody library (L006 4A8/CC12.1/COV2-2489) was generated using Gibson Assembly with the gel extracted components in a 2:1 molar ratio of antibody library insert to pBMP103-UMI yeast surface display vector. Each of the antibody combinatorial libraries contained 64 variants and were mixed in an equimolar amount for the Gibson Assembly reaction[28] using the NEBuilder HiFi DNA Assembly master mix following the manufacturer's protocol. A column clean-up was performed to remove residual buffer and enzymes and approx. 25% of the cleaned reaction product was transformed into homemade chemically competent *E. coli* Mach1. The next day, 11,000 transformants were observed from the transformation dilution plate and the entire library was harvested and miniprepped.

The barcoded yeast surface display libraries for the S1/HA mixed antigen sort were generated by Gibson assembly of the gel extracted components in a 2:1 molar ratio of antibody library insert to yeast surface display vector for the kappa and lambda antibody libraries separately. L024 and L029 were assembled with L018 library and L030, OMIL004, OMIL006, OMIL011, and OMIL0012 were assembled with pBMP103-UMI library in a total reaction volume of 20 µL. Both Gibson Assembly reactions were incubated for 4 hours at 50 °C and the reaction products were cleaned and concentrated to 6 µL with a Monarch DNA & PCR Cleanup Kit (NEB). The entire 6 µL product was transformed via electroporation into TransforMAX cells (Lucigen, EC300110) and incubated at 37 °C overnight. A dilution plate was used to assess the transformation efficiencies and the transformants were bottlenecked to 2000 lambda variants and approximately 8000 kappa variants. The bottlenecked libraries were grown up in 50 mL SOB + kanamycin overnight at 37 °C. The next day the libraries were miniprepped and pooled together and 6 different 4A8 barcoded plasmids were also spiked into the library pool. 5 µg of plasmid DNA was transformed into chemically competent *Saccharomyces cerevisiae*

(EBY100, ATCC MYA-4941) in parallel reactions for the biological replicate libraries and stored as yeast glycerol stocks in -80 °C according to Medina-Cucurella & Whitehead[64].

## Barcode-variant pairing

Barcodes were paired with $V_H$ and $V_L$ variants through Oxford nanopore sequencing or by short-read sequencing of amplicons prepared by intramolecular ligation of barcode in proximity to the CDR3 of either the $V_H$ or $V_L$ using Golden Gate[27]. Oxford nanopore sequencing (Plasmidsaurus) was performed on individual plasmids. Short-read amplicons were sequenced on an Illumina MiSeq with 2×250 paired end reads (Rush University Sequencing Core). For intramolecular ligation, two replicates were performed independently.

The optimized intramolecular ligation procedures were performed in a reaction volume of 100 µL with 1 µg of plasmid library, 400 U of T4 DNA ligase (1 µL of 400 U/µL), and 1X T4 DNA ligase buffer (NEB; catalog # B0202S). Also added to the reaction were either 30 U of BbsI (3 µL of 10 U/µL) (NEB; catalog # R0539L) - for the intramolecular ligation of barcode to VL - or 30 U of PaqCI (3 µL of 10 U/µL) plus 3 µL PaqCI activator (diluted from 20 µM stock 1:4 in 1X T4 DNA ligase buffer) (NEB; catalog # R0745L) - for ligation of barcode to VH. The reaction was subjected to 60 cycles of 37 °C for one minute followed by 16 °C for one minute, and then a final incubation of 37 °C for 5 minutes. Exonuclease III was added to each reaction (1 µL of a 1:10 dilution made in 1X rCutSmart buffer from 100 U/µL stock) (NEB; catalog # M0206L) followed by incubation at 37 °C for 30 minutes. We performed electrophoresis on the reactions on 1% (w/v) agarose gels in 1X TAE and gel extracted (Macherey-Nagel, catalog # 740609.50) the bands corresponding to intramolecular ligated products for the VH-barcode and VL-barcode pairing.

Amplicons were prepared by first performing a PCR using primers 428 - 431 to amplify the barcode and gene sequence (UMI-VL: 428 & 430; UMI-VH: 429 & 431) and append Illumina TruSeq small RNA compatible sequences. 10 ng of gel extracted input DNA was amplified in a 25 µL total reaction with Phusion High-Fidelity Polymerase with reaction components following the manufacturers recommended protocol. The PCR thermocycler program progressed for 12 cycles. After the first PCR was completed, a shrimp alkaline phosphatase (rSAP) clean-up step was performed according to the manufacturer's instructions on 10 µL of PCR product. 1 µL of rSAP cleaned DNA was used as the input for the seconod PCR, which amplifies the amplicon further and appends unique 6-bp TruSeq small RNA barcodes and Illumina sequencing adapters. The second PCR was performed using Phusion High-Fidelity Polymerase with reaction components following the manufacturers recommended protocol for 14 cycles with a 25 µL total reaction volume. After the second PCR, the amplicons were cleaned with Ampure XP beads (Beckman Coulter, A63880) following the manufacturer's instructions. Each dsDNA PCR product was quantified using Quant-IT PicoGreen (Invitrogen, P11495) and pooled for deep sequencing.

For optimization of the protocol above, three individual plasmids were sequenced by Oxford nanopore. These plasmids were then mixed in pre-defined ratios and different amplicon prep conditions were applied. These differences include: the polymerase used, the number of PCR cycles, and the type of PCR clean-up between the first and second PCRs. The three polymerases used were Phusion High-Fidelity DNA polymerase (NEB, M0530), Q5 polymerase in a 2X Master Mix (NEB, M0492), and KAPA polymerase (Roche, 7958927001) and each PCR reaction components and thermocycler program were performed according to the manufacturer's instruction.

## Yeast Cell Surface Titrations

To determine the binding affinity of individual variants, isogenic titrations were performed according to Chao et al.[65]. A population of yeast displaying a single Fab variant are induced to display copies of

the Fab on the surface and different concentrations of antigen are added in separate reactions. Fluorescence tags as a readout for antigen binding are analyzed using flow cytometry and fractional saturation curves of the titrated antigen concentration compared to fluorescence intensity allow for determination of the variant monovalent binding affinity to the antigen. 4A8 variants were made by the method of combinatorial nicking mutagenesis[32]. Each variant was tested in duplicate on two separate days (n = 4 total replicates) and compared with a titration of mature 4A8 Fab to determine the log $K_D$ ratio ($\log_{10}(K_D/K_{D,wt})$). The isogenic titrations reported in Fig. 1 were reported in at least duplicate (n ≥ 2).

## Sorting of Fab libraries

For sorting the mixed 4A8/COV2-2489/CC12.1 library, 1e7 (ten million) yeast library cells from glycerol stocks were shaken at 230 rpm and grown in 250 mL flasks at 30 °C overnight in 50 mL SDCAA + PenStrep and kanamycin. The next day, the 1e7 yeast cells were induced in SGDCAA + PenStrep and kanamycin at 20 °C for 48 hours in a total reaction volume of 50 mL. On the morning of sorting the cells were concentrated to an $OD_{600}$ = 5 in ice-cold PBSF. Ten million library cells were then labeled with different amounts of S1-hFc-His at the following concentrations in nM: 0, 1, 2.5, 5, 10, 50, 100, 250, 500, 1000, 2000 for 30 minutes at room temperature. After the binding reactions were finished cells were spun down, washed with 1 mL of ice-cold PBSF, and then labeled with 6.25 µL anti-V5-AlexaFluor488 (ThermoFisher, 37-7500-A488, 1 mg/mL) and 25 µL Goat anti-hFc-PE (ThermoFisher, 12-4998-82, 0.5 mg/mL) for 30 minutes covered on ice. After fluorophore labeling, the cells were pelleted and washed with 1 mL of ice-cold PBSF, and pellets were left covered on ice until loading onto Sony SH800 cell sorter, at which time each pellet was resuspended in 5 mL of ice-cold PBSF. Cells were first gated for yeast cells and single cells (drawn according to Banach et al.[66] to avoid collection of clumped yeast of irregular large yeast aggregates), and then a gate for positive Fab expression was drawn and 200,000 cells were collected as the library reference population (Supplementary Fig. 6). Sorting bins for the Top 25% and Next 25% of binding based on PE signal were gated from the display positive population and 200,000 cells were collected in each bin (Supplementary Fig. 6). Sorted cells were recovered in 1 mL of SDCAA plus antibiotics overnight at 30 °C, at which time another 1 mL of SDCAA was added. Cells were grown until they reached an $OD_{600}$ greater than 2. Cell stocks were made for each sorted population at 1 mL of $OD_{600}$ = 1 in yeast storage buffer (20% w/v glycerol, 20 mM HEPES-NaOH, 200 mM NaCl, pH = 7.5).

For sorting the S1/HA library, 1e7 (ten million) yeast library cells from glycerol stocks were shaken at 230 rpm and grown in 250 mL flasks at 30 °C overnight in 50 mL SDCAA + PenStrep and kanamycin. The next day, the 1e7 yeast cells were induced in SGDCAA + PenStrep and kanamycin at 20 °C for 48 hours in a total reaction volume of 50 mL. On the morning of sorting the cells were concentrated to an $OD_{600}$ = 5 in ice-cold PBSF. Ten million library cells were then labeled with different amounts of S1-hFc-His and biotinylated HA for 30 minutes at room temperature. The 11 labeling concentrations spanned from 2.5 nM – 2000 nM and included mixes of both S1-hFc-His and biotinylated HA. After the binding reactions were finished cells were spun down, washed with 1 mL of ice-cold PBSF, and then labeled with 6.25 µL anti-V5-AlexaFluor488 (ThermoFisher, 37-7500-A488, 1 mg/mL, Lot#:YD372483), 25 µL Goat anti-hFc-PE (ThermoFisher, 12-4998-82, 0.5 mg/mL, Lot#:2626356), and 25 µL SAPE (ThermoFisher, S866) for 30 minutes covered on ice. After fluorophore labeling, the cells were pelleted and washed with 1 mL of ice-cold PBSF, and pellets were left covered on ice until loading onto Sony SH800 cell sorter. Each pellet was resuspended in 5 mL of ice-cold PBSF and loaded on to the cell sorter. Cells were first gated for yeast cells and single cells, and then a gate for positive Fab expression was drawn and 1,000,000 cells were collected per bin for the first replicate and 750,000 cells per bin

were collected for the second replicate. Sorted cells were recovered in 5 mL of SDCAA plus antibiotics for at least 30 hours at 30 °C and cell stocks were made for each sorted population in yeast storage buffer. Yeast biological replicates were performed. The plasmid encoded master library was prepared once and separately transformed into yeast; these libraries were sorted on separate days.

## Amplicon preparation and deep sequencing

Plasmid DNA from each collected population (1e6–4e6 sorted yeast cells) was prepared according to Medina-Cucurella & Whitehead[64] using Zymoprep Yeast Plasmid Miniprep kits in either individual Eppendorf tubes (D2004) or 96-well plate format (D2007) and plasmid DNA was eluted in 30 µL nuclease free water. 15 µL of eluted plasmid DNA was further purified with exonuclease I and lambda exonuclease. The barcode region of the purified DNA was amplified using 25 PCR cycles with Illumina TruSeq small RNA primers following Kowalsky et al. 'Method B'[67] using Phusion High-Fidelity DNA polymerase in a 50 µL total reaction. 5 µL of the PCR product was size verified on a 1% (w/v) agarose gel and the remaining 45 µL was cleaned with Ampure XP beads (Beckman Coulter, A63880) following the manufacturer's instructions. Each dsDNA PCR product was quantified using Quant-IT PicoGreen (Invitrogen, P11495) and pooled in equimolar amounts for deep sequencing. Amplicons were sequenced on either an Illumina MiSeq (4A8/CC12.1/COV2-2489 sort) or NovaSeq6000 (S1/HA sorts) by Rush University with single end reads.

## Data processing

Sequencing files were processed using the custom Python code accessible on GitHub (https://github.com/WhiteheadGroup/MAGMA-seq). The code contains three primary modules used in this work referred to as haplotyping, scanning, and parameter estimation. Haplotyping takes input sequencing files from internally ligated yeast display plasmids and creates a barcode-to-variant map. Scanning reads input sequencing files for sorted yeast populations for which only barcode sequences are processed, counts, and matches the barcodes to a variant specified in the previously generated barcode-to-variant map, and integrates this with sorting conditions for final output. Finally, the parameter estimation module performs maximum likelihood estimation (MLE) on each variant contained in the output to generate parameter estimates for $K_D$ and $F_{max}$ with 95% confidence intervals determined from reduced chi squared. See the "config" folder on GitHub for exact parameters used for generating each of the datasets used in this work.

For each of the sequencing processing modules (haplotyping and scanning), FASTQ files and processing parameters are entered into a config file (see README and example config files on GitHub repository). Necessary packages including Biopython, NumPy, and SciPy can be easily installed into a conda virtual environment with the included YAML file. The code is highly efficient and parallelizable (using the multiprocessing library) and can run on datasets containing millions of sequences in under an hour on our hardware (Alpine supercomputing cluster (CU Research Computing) x86_64 AMD Milan CPU with 32MB L3 Cache (utilizing 8 cores), 3.75 GB RAM/core).

## Sequence merging and filtering

We adapted the software from Haas et al.[68] for merging paired end reads at all the pertinent amplicon lengths. Sequences are then filtered based on the sequence agreement within overlap regions (see Haas et al.[68] for algorithm details) as well as overall minimum quality scores across the full amplicon, gene, and barcode regions. Barcodes and genes are extracted from these successfully merged reads assuming a fixed location within the amplicon. Sequences are then collapsed and counted based on unique barcode and gene combinations and gene sequences are matched to a set of possible wild-type sequences based on Hamming distance. Amino acid mutations are then determined

based on the chosen wild-type sequence. Variant frequencies are calculated by considering the genotypes represented in the dataset, ignoring barcodes.

A six-letter sequence motif (CGGCGG) occurring within the COV2-2489 antibody VH gene causes a precipitous drop in quality scores of all base calls downstream from this previously known motif[69] on the reverse read (Supplementary Fig. 5). This drop in quality score required a different haplotyping protocol for this antibody library where paired reads were not merged. This is justifiable as the mutations encoded in our library all exist on the high quality forward read, and the barcodes are located on the reverse read upstream of the quality drop. We identified reads from this antibody by matching the read to an unmutated region at the beginning of the sequence (VH positions 1–31), filtered the individual forward reads based on quality (dropping reads with overall minimum quality <Q10), and then paired the associated barcode from the reverse read based on index. The resulting data was passed into the following haplotyping step identical to the other merged reads.

### Haplotyping: pairing a barcode with Fab variant

For each barcode, we make a variant call by comparing all observed barcode-genotype pairings. We first apply a mutation filter (variants with more than 10 mutations from the assigned wild-type are removed). We then apply the count filter and a frequency filter (see config for filtering values used). Additionally, options are available for removing all silent mutations and/or all mutations not encoded in the mutagenic library. From the remaining possible pairings, we divide the observed read counts by the variant frequency and select the variant with the maximum of these values. This method appropriately weights lower observed counts of rarer variants. The raw pairing data as well as the processed barcode-to-variant map are output to CSV files for analysis.

After barcode maps have been made for VH and VL segments, we merge the two maps based on identical barcodes and output the resulting pairings as a CSV file. A final check is performed where barcodes that pair to heavy and light chains from distinct antibodies are removed from the map, resulting in a barcode-to-variant map.

### Scanning

The scanning module matches sequenced barcodes from sorted libraries to barcodes in the map produced by the haplotyping module. It is designed to process single reads only (alternatively, forward reads from paired-end sequencing runs can be used). As described previously, barcodes are identified based on fixed location within the read. As a default, barcodes are filtered based on adherence to the template sequence of mixed bases; this option can be turned off. Each barcode is matched to an identical barcode in the barcode-to-variant map and the number of times each barcode is seen is summed. Information entered by the user in "limit.csv" including high and low bin limits (Hj and Gj, respectively), and number of cells sorted and collected in each bin (Nk and Njk, respectively) are matched with concentration and bin names to generate the CSV file needed for the next parameter estimation step. Note that the concentration and bin names in the "limit.csv" must match the identifiers from the scanning barcodes config file. For example, a bin named "top25" at concentration "5 nM" should be identified in the config file as "conc5nM_bintop25".

Scan barcode outputs information for each of the concentrations and bins analyzed as well as an overall output in two forms: (1) "_combined.csv" records the counts by barcode and (2) "_collapsed.csv" records the counts collapsed by antibody variant. Both the combined and collapsed CSV files are ready for input into the parameter estimation module assuming a proper "limit.csv" was specified. For each bin and concentration, the percentage of barcodes that matched to a variant in the map is recorded. With our conservatively filtered maps, these percentages tend to range between 50 and 70%

read matching. Far lower percentages usually indicate low efficiency of haplotyping.

### Parameter Estimation

A complete description of the mathematics behind parameter estimation is detailed in Supplementary Note 1 (Supplementary Figs. 11-13, Supplementary Table 1). Custom Python software was used to estimate variant-specific monovalent binding dissociation constants ($K_{d,i}$) and mean maximum fluorescence at saturation ($F_{max,i}$) fit by Eq. (1). These values were inferred using maximum likelihood estimation of the following expression for the log likelihood $LL_i(K_{d,i}, F_{max,i})$:

$$LL_i(K_{d,i}, F_{max,i}) = -\sum_{jk}\left(\frac{p_{ijk} - Model_{ijk}}{\sigma_{ijk}}\right)^2 \qquad (2)$$

Here, $p_{ijk}$ is the probability of capturing variant i in bin j at labeling concentration k and is determined from observables from the deep sequencing experiment according to the following equation:

$$p_{ijk} = \varnothing \frac{\frac{r_{ijk}}{\sum_i r_{ijk}}}{\frac{r_{ir}}{\sum_i r_{ir}}} \qquad (3)$$

$\varnothing$ is the total fraction of cells collected in the sorting bin relative to the reference sample, $r_{ijk}$ is the number of observed read counts for variant i in bin j at labeling concentration k, $r_{ir}$ is the number of observed read counts for variant i in the reference population, and the summations represent the sum of observed read counts over all barcodes.

$Model_{ijk}$ is the model probability of the variant i sorting in bin j at labeling concentration k and is defined as:

$$Model_{ijk} = \frac{1}{2}erf\left(\frac{lnF_{gjk} - lnF_{ik} + \frac{1}{2}\sigma^2}{\sigma\sqrt{2}}\right) - \frac{1}{2}erf\left(\frac{lnF_{g2jk} - lnF_{ik} + \frac{1}{2}\sigma^2}{\sigma\sqrt{2}}\right) \qquad (4)$$

Here, $F_{gjk}$ and $F_{g2jk}$ are the gating boundaries in the selected bin j, and $\sigma$ is the standard deviation of the log normal distribution and set to 1.02 for all variants. Different parameter values in Eq. (1) change the variant-specific mean fluorescence $F_{ik}$ at each labeling concentration used in the experiment.

The parameter $\sigma_{ijk}$ representing the uncertainty in the probability of sorting is defined as:

$$\sigma_{ijk} = \sqrt{\left(\sigma_{ijk,extrinsic}\right)^2 + p_{ijk}^2\left(\frac{1}{r_{ijk}} + \frac{1}{r_{ir}}\right)} \qquad (5)$$

For sorts reported in Fig. 2, $\sigma_{ijk,extrinsic}$ was set to 0.02. For the sorts reported in Fig. 4, this value was measured using the average probabilities of the non-binding mutants of antibodies 1G01 and 1G04.

Parameter estimation requires that the data be supplied in the format of the scanning module output (see examples/scan_output.csv for example) where each row specifies a single observation of a variant labeled at a given concentration and collected at a given bin. Additionally, a few global parameters must be entered by the user. First, sigma defines the width of the lognormal distribution that represents the possible fluorescence range for a given variant. We have found that this value is somewhat independent of the variant and label concentration. In our testing, these values range from 0.90 to 1.02. Second, "B" represents the variant-independent cell autofluorescence, which can be determined by reading fluorescence values of Fab-expressing yeast cells without binding partner. We find that this value should fall in the range of 290–350 RFU (in PE channel) using our

equipment (Sony SH800, yeast cells, 488 nm laser with compensation for PE/AlexaFluor488 fluorophores).

To achieve accurate estimates for weak binders and poorly expressed Fabs (high $K_D$ or low $F_{max}$) a few modifications to the MLE algorithm were necessary. First, manual curation was used to remove bins that had poor sequencing coverage or that had inferred probabilities that were inconsistent with the other datasets. For the data represented in Figs. 2 and 3, the two bins affected were at the 25 nM labeling concentrations. The MLE algorithm first performs parameter estimation using all remaining top 25% bins. The maximum likelihood estimates were analyzed and variants with calculated $K_D$ that exceeded 1 µM or $F_{max}$ that fell under 12,000 RFU were removed. For these variants, MLE was performed again by concatenating top 25% and next 25% bins into a single top 50% bin at each concentration using the joint probability estimate using equation (7). For the mixed antigen sorts represented in Figs. 4 and 5, 2–7 variants were assessed by combining both bins for 25 and 50 nM labeling concentrations. Additionally, all anti-S1 probabilities $p_{ijk}$ were multiplied by 0.64 to correct for cell sorter efficiency in this experiment.

### Supervised learning

Programmed mutations for reverse trajectory libraries were one-hot encoded using the custom python notebook One-hot-encode.ipynb. Ordinary least squares (OLS), least absolute shrinkage and selection operator (LASSO), and ridge regression analyses were performed on the one-hot encoded variants for $\log(K_{D,i}/K_{D,WT})$ (4A8 titrations and MLE) and $F_{max}$ (MLE) regularization using custom Python Jupyter notebooks OLS.ipynb, LASSO.ipynb, and Ridge.ipynb. Coefficient weights and error values for each regression technique and model order are detailed in Supplementary Data 3.

Sequences of anchor mAbs used in this study are from Guthmiller et al.[44] Clonal analyses were performed using VGenes (https://wilsonimmunologylab.github.io/VGenes/) using sequences from Guthmiller et al.

### Statistics & reproducibility

In all experiments in this study, no data were excluded from the analysis. Yeast libraries were prepared in independent replicates and sequenced at high depth of coverage to confirm $K_D$ estimates. All statistical tests were computed either using SciPy or custom code and are included in the Python scripts freely available on GitHub.

### Reporting summary

Further information on research design is available in the Nature Portfolio Reporting Summary linked to this article.

## Data availability

Deep sequencing data is available on the sequencing read archive under SRA accession PRJNA1026152, accession codes SRR26328231, SRR26328232, and SRR26328233. The plasmids for constructing compatible workflow Fabs pBDP (AddGene ID: 217827), pMMP_kappa (AddGene ID: 217828), pMMP_lambda (AddGene ID: 217829), pYSD_kappa_mRFP (AddGene ID: 217830), and pYSD_lambda_mRFP (AddGene ID: 217831), as well as positive control plasmids p4A8_S7T_BC (AddGene ID: 217832) and p4A8_M59I_T94M_BC (AddGene ID: 217833), are freely available from AddGene (Deposit: 84029). Source data are provided with this paper.

## Code availability

All custom scripts and code are freely available on GitHub (https://github.com/WhiteheadGroup/MAGMA-seq).

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

## Acknowledgements

This work was supported by the National Institute Of Allergy And Infectious Diseases of the National Institutes of Health (Award Numbers 5R01AI141452-05 to T.A.W.; R00AI159136 to J.J.G.), the US Department of Education (Award Number P200A180034, participant support B.M.P.), the National Science Foundation Graduate Research Fellowship Program (Z.T.B. DGE Award Number 2040434, fellow ID: 2021324468), the NSF REU (Award #2244288 for K.M.C.), and the NIH/CU Molecular Biophysics Program. This work utilized the Alpine high performance computing resource at the University of Colorado Boulder. Alpine is jointly funded by the University of Colorado Boulder, the University of Colorado Anschutz, Colorado State University, and the National Science Foundation (award 2201538). The authors also acknowledge Dan Schwartz for useful discussions around MLE, Pete Tessier around barcode tagging, John Jumper for helping coin the term 'wide mutational scanning', and Kevin Kunstman, Cecilia Chau, and Ashley Wu at Rush University for helpful discussions around NGS.

## Author contributions

Conceptualization: B.M.P., M.B.K., T.A.W. Designed plasmid sets: B.M.P., M.B.K., O.M.I., Z.T.B., E.R.R., T.A.W., Designed bench research: B.M.P., M.B.K., I.S., T.A.W. Performed bench research: B.M.P., M.B.K., O.M.I., I.S., C.M.H., A.M.W., S.A.U. Developed computational algorithms: B.M.P., M.B.K., K.M.C., P.J.S., T.A.W. Developed code: B.M.P., M.B.K., K.M.C., P.J.S. Data analysis: B.M.P., M.B.K., O.M.I., J.J.G., T.A.W. Contributed reagents: E.A., J.J.G. Writing: B.M.P., M.B.K., O.M.I., T.A.W. Supervision: T.A.W. Funding Acquisition: T.A.W., J.J.G., and Z.T.B.

## Competing interests

The authors declare no competing interests.
