## [Peer Review File · Nature Communications]

REVIEWER COMMENTS

Reviewer #1 (Remarks to the Author):

The paper describes the development and validation of a technology called MAGMA-Seq for extracting binding data for Fab antibody mutant libraries. The technology is an extension (and improvement) of an earlier method (Tite-Seq) but more useful as it involves Fab expression rather than scFv in Tite-Seq and incorporates a number of useful features. A number of examples of antigen binding analyses of small libraries of Fabs using MAGMA-Seq are presented. While the authors should be commended for the large amount of work described in the paper there are some major issues that need to be addressed before further consideration for NC.

First and foremost the authors attempt to extract DDG values from the analysis of barcode Seq counts following FACS of yeast displayed Fab libraries. This is not appropriate. DDG is a clearly defined physicochemical quantity and $\Delta\Delta G$ values must be determined by precise measurements that quantify accurately the amount of antibody:antigen complex etc. YSD can be used to determine RELATIVE Kd values but these values are not actual KDs and hence extracting DDG values from such data can be misleading.

Second, but related, the YSD Kd values shown in Figure 1C do not seem consistent with published and properly measured values in vitro using SPR or related techniques. For example the published KD for 4A8 is <1 nM ie more than 20 fold lower than the value measured by YSD. I noticed similar discrepancies with published values for other antibodies on the table. Along these lines not sure why 1G01 and 1G04 showed sub-micro molar binding to NA? What exactly is NA RBS listed in 1C?

Third, when yeast cells are expressing Fabs that bind to different epitopes on one multivalent epitope binding of cells expressing one Fab could lead to antigen clustering which would then favor high avidity interactions with a different yeast cell expressing an antibody for the second epitope. Do the authors see cell doublets or aggregates in the FACS sorts and if so, have they examined how they affect the measured Kd values in such experiments?

Minor: Better to avoid terms such as "haplotyping" which has a very specific meaning in genetics. Similarly "isogenic titrations" should at least be defined.

Paragraph 298-309 is confusing.

Reviewer #2 (Remarks to the Author):

Petersen and Kirby et al. present MAGMA-seq, a biotechnology that enables simultaneous quantitative assessment of the binding of multiple antibody variants against multiple antigens. While other techniques exist that use surface display to perform deep mutational scanning of individual binding contexts, MAGMA-seq is distinguished by its molecular barcoding approach that allows variants of antibodies and different antigens to be explored in the same experiment, which the authors highlight is crucial to provide the data regimen required for deep learning methods to learn more generalisable patterns of molecular recognition rather than biases associated with any one binding site (e.g. Hummer et al. 2023, 10.1101/2023.05.17.541222).

Here, the authors first benchmark MAGMA-seq's ability to quantitatively estimate binding affinity, through a maximum likelihood estimation based on the number of reads observed for a variant at a given fluorescence bin and gating threshold. This methodology is clearly explained in the supplementary note and produces good agreement with isogenic titration values when there is sufficient sequencing coverage. They then proceeded to explore the binding of antibody variants to multiple antigens (two antigens, six epitopes), and show that even exploring a relatively small percentage of potential back-mutations (c. 3%) to the inferred unmutated common ancestor (UCA) recapitulates plausible development landscapes that can be rationalised structurally. Additionally, they showed evidence that MAGMA-seq can be applied prospectively by starting from UCA_2-17 and exploring single nucleotide substitutions in the framework and CDRs, yielding a few candidate antibodies with up to an estimated 4x stronger binding affinity. Finally, their Fab libraries were analysed to derive binding rules for influenza antibodies that were consistent with the spread of V gene origins seen amongst lineages known to bind the same site.

As presented, this approach would be of immediate interest to computational researchers designing clustering approaches to group antibodies with common functional specificities and is a key development en route to the volume of data required needed to train generalisable machine learning algorithms. Limitations of the technology are thoroughly covered in the discussion section, amongst which a primary factor is that yeast display constrains MAGMA-seq to the study of binders in the nanomolar range, limiting its ability to guide engineering towards sub-nanomolar affinities often seen in therapeutic antibodies or to provide a baseline for weak binders (e.g. COV2-2489).

Overall, the work is timely and has been executed to a high level of rigour. I have only a few comments:

1. Please could the authors comment on the somatic hypermutation level of their antibodies? Many anti-SARS-CoV-2 antibodies are remarkably germline and so highly similar in sequence to their UCA, which may account for their sparse development profiles. Did they find that mature antibodies with a larger number of accrued mutations away from the UCA exhibited more complex development profiles?

2. When applying MAGMA-seq in a prospective sense, the authors used a "subset" of a library containing all possible single-point framework and CDR mutations from a high nanomolar binder (UCA_2-17).

(a) What fraction of the theoretical library was explored, and were these rationally or randomly chosen?

Then, in the next section, they mention "single and double mutants" of the 222-1C06 CDRH2.

(b) Were the 222-1C06/319-345 influenza bnAb libraries made using the same mature/UCA chimera strategy as the SARS-CoV-2 binders? If so, it may be clearer to present these two results in a different order or using a different linking phrase on page 7 line 298. Otherwise, if they were made using a different approach, please could the authors clarify what library construction strategy they used?

3. Please could the authors comment on whether there are any limitations on the diversity of epitopes that can be studied in a single experiment? For example, if an antibody lineage's binding mode to an epitope sterically overlaps with another antibody lineage's mode to a nearby epitope, does this not risk distorting the fluorescence signal for the weaker binder (via competitive inhibition)?

Minor:

Page 3, line 120: in this context, I suggest changing "epitopes" to "domains"; there are several classes of epitope within the RBD (10.1016/j.cell.2021.02.032).

Page 4, line 184: biophysical -> biophysically

Page 6, line 270: Should S74F be in this set?

Page 8, line 346: "four variants" but only three mutations listed

Reviewer Responses

Reviewer #1 (Remarks to the Author):

The paper describes the development and validation of a technology called MAGMA-Seq for extracting binding data for Fab antibody mutant libraries. The technology is an extension (and improvement) of an earlier method (Tite-Seq) but more useful as it involves Fab expression rather than scFv in Tite-Seq and incorporates a number of useful features. A number of examples of antigen binding analyses of small libraries of Fabs using MAGMA-Seq are presented. While the authors should be commended for the large amount of work described in the paper there are some major issues that need to be addressed before further consideration for NC.

First and foremost the authors attempt to extract DDG values from the analysis of barcode Seq counts following FACS of yeast displayed Fab libraries. This is not appropriate. DDG is a clearly defined physicochemical quantity and $\tau\theta\epsilon$ values must be determined by precise measurements that quantify accurately the amount of antibody:antigen complex etc. YSD can be used to determine RELATIVE K_d values but these values are not actual K_ds and hence extracting DDG values from such data can be misleading.

We rewrote all sections in the main text and figures replacing all DDG values as $\log_{10}(K_{d,i}/K_{d,wt})$, where $K_{d,i}$ is the MLE yeast dissociation constant for variant i , and $K_{d,wt}$ is the MLE yeast dissociation constant for the antibody parental background.

Second, but related, the YSD K_d values shown in Figure 1C do not seem consistent with published and properly measured values *in vitro* using SPR or related techniques. For example the published K_d for 4A8 is <1 nM ie more than 20 fold lower than the value measured by YSD. I noticed similar discrepancies with published values for other antibodies on the table. Along these lines not sure why 1G01 and 1G04 showed sub-micro molar binding to NA? What exactly is NA RBS listed in 1C?

There are two broader points raised here, which we address.

The first point relates to a direct comparison between *in vitro* results from SPR/ELISAs and yeast measurements. The major difficulty in applying such comparisons is that many papers reporting *in vitro* measurements do not ensure *monovalent* binding. Most reported dissociation constants are reported with antibodies in an IgG bivalent format, and viral glycoproteins chosen are often trimeric or are otherwise deposited at high enough surface densities to promote multimerization. For example, the Chi et al Science 2020 paper reports a 0.99 ± 0.05 nM K_d for 4A8, but in the methods this was measured using full Spike *trimer*. Thus, the K_d reported may be an effective K_d combining monovalent binding interactions with avidity effects involving binding of the bivalent IgG across protomers. Consistent with this, the same paper also reported a K_d 92 ± 0.05 nM

for 4A8 when assayed against S1 (monomer), which is more consistent with the results presented here. The monovalent K_d values we observe for 319-345, 222-1C06 (Guthmiller et al. Nature 2022) and CR6261 (Throsby et al. PLoS One 2008) are consistent within a factor of 4 for related HA H1 trimers, which may reflect the inability of these antibodies to bind bivalently. With respect to the 1G01/1G04 antibodies, we re-analyzed the data and realized the neuraminidase used in the experiments was bound by other antibodies non-specifically. Because we were not confident in the quality of the neuraminidase antigen used, we thought it safest to remove all results for 1G01/1G04 from analysis (Figure 1C).

The second point is direct comparison of yeast display measured monovalent dissociation constants vs. gold standard *in vitro* measurements like SPR. SPR is a gold standard for a reason, and we certainly do not wish to claim that yeast-derived measurements are superior. We have added a sentence in the discussion with this limitation (lines 431-433):

“ Fifth, we note that, due to the implementation of FACS with yeast display, accuracy of MAGMA-seq estimated binding affinities may not precisely match gold-standard *in vitro* measurements like SPR, where antibody/antigen interactions are more directly quantified. “

Third, when yeast cells are expressing Fabs that bind to different epitopes on one multivalent epitope binding of cells expressing one Fab could lead to antigen clustering which would then favor high avidity interactions with a different yeast cell expressing an antibody for the second epitope. Do the authors see cell doublets or aggregates in the FACS sorts and if so, have they examined how they affect the measured K_d values in such experiments?

This is an excellent technical point. We control for this possibility in our implementation. First, we minimize cell aggregates by (i.) optimization of our yeast growth and induction protocols; and (ii.) by ensuring a high molar ratio of antigen to total displayed protein (between 10- to 100-fold molar excess). Second, we explicitly gated on single cells (no doublets/aggregates) in our demonstrations in this paper. Our full gating strategy is shown in Extended Data Figure 6 (see first row, second column stating ‘single cells’). To discriminate single cells from cell aggregates we employ an additional light scattering gate (FSC-H vs. FSC-A) first described in Banach et al J Exp Medicine 2022 from Brandon Dekosky’s group at MIT. The Sony cell sorter we use (SH800) uses different parameters for the forward scatter, for which it is possible to gate small single and budding cells away from clumped cell aggregates. Clumped/aggregated cells fall below the FSC-H/FSC-A diagonal. We have now included this reference and explained this in the methods (lines 512-513).

Minor:

Better to avoid terms such as "haplotyping" which has a very specific meaning in genetics. Similarly "isogenic titrations" should at least be defined.

To avoid confusion of these terms we have replaced references containing the term “haplotyping” with “barcode pairing”. Additionally, we have inserted a definition of “isogenic titrations” in lines 70-71.

Paragraph 298-309 is confusing.

We concur. Reviewer #2 made a similar point. To address this, we added an additional explanatory paragraph (lines 320-325).

“The libraries described thus far are all retrospective analyses of antibody development trajectories, where libraries encoded chimeras of the mature and UCA sequences. **To further investigate the utility of this method, our second demonstration of MAGMA-seq included a prospective antibody development library and a few CDR targeted site-saturation mutagenesis libraries. We generated each of these antibody libraries in parallel reactions and subsequently pooled and barcoded the variants. We bottlenecked the library, which selected individual variants randomly, and assessed it with MAGMA-seq.**”

Reviewer #2 (Remarks to the Author):

Petersen and Kirby et al. present MAGMA-seq, a biotechnology that enables simultaneous quantitative assessment of the binding of multiple antibody variants against multiple antigens. While other techniques exist that use surface display to perform deep mutational scanning of individual binding contexts, MAGMA-seq is distinguished by its molecular barcoding approach that allows variants of antibodies and different antigens to be explored in the same experiment, which the authors highlight is crucial to provide the data regimen required for deep learning methods to learn more generalisable patterns of molecular recognition rather than biases associated with any one binding site (e.g. Hummer et al. 2023, 10.1101/2023.05.17.541222).

Here, the authors first benchmark MAGMA-seq's ability to quantitatively estimate binding affinity, through a maximum likelihood estimation based on the number of reads observed for a variant at a given fluorescence bin and gating threshold. This methodology is clearly explained in the supplementary note and produces good agreement with isogenic titration values when there is sufficient sequencing coverage. They then proceeded to explore the binding of antibody variants to multiple antigens (two antigens, six epitopes), and show that even exploring a relatively small percentage of potential back-mutations (c. 3%) to the inferred unmutated common ancestor (UCA) recapitulates plausible development landscapes that can be rationalised structurally. Additionally, they showed evidence that MAGMA-seq can be applied prospectively by starting from UCA_2-17 and exploring single nucleotide substitutions in the framework and CDRs, yielding a few candidate antibodies with up to an estimated 4x stronger binding affinity. Finally, their Fab libraries were analysed to derive binding rules for influenza antibodies that were consistent with the spread of V gene origins seen amongst lineages known to bind the same site.

As presented, this approach would be of immediate interest to computational researchers designing clustering approaches to group antibodies with common functional specificities and is a key development en route to the volume of data required needed to train generalisable machine learning algorithms. Limitations of the technology are thoroughly covered in the discussion section, amongst which a primary factor is that yeast display constrains MAGMA-seq to the study of binders in the nanomolar range, limiting its ability to guide engineering towards sub-nanomolar affinities often seen in therapeutic antibodies or to provide a baseline for weak binders (e.g. COV2-2489).

Overall, the work is timely and has been executed to a high level of rigour. I have only a few comments:

1. Please could the authors comment on the somatic hypermutation level of their antibodies? Many anti-SARS-CoV-2 antibodies are remarkably germline and so highly similar in sequence to their UCA, which may account for their sparse development profiles. Did they find that mature antibodies with a larger number of accrued mutations away from the UCA exhibited more complex development profiles?

We agree that the SARS-CoV-2 antibodies chosen for analysis (4A8, CC12.1, 2-7) have low levels of SHM, and this may impact our findings. There has been limited work on development trajectories of antibodies with a larger number of accrued mutations. In this work, we tested the influenza antibody CR6261 containing many more mutations. and found similar sparse development profiles. Tite-seq was previously used to evaluate CR6261 and a related antibody CR9114 (albeit as scFvs). A recent re-analysis of these datasets by the Thornton group (<https://www.biorxiv.org/content/10.1101/2023.09.02.556057v1.full>.) supports sparse development of these antibodies, as assessed by yeast measurements. However, the number of antibodies tested (large vs. small number of accrued mutations) is too small for statistical comparison. Ultimately, the answer to the above intriguing question will come by evaluating a larger number of antibody trajectories. We have included a reference to the above Thornton paper in the discussion (line 397).

2. When applying MAGMA-seq in a prospective sense, the authors used a "subset" of a library containing all possible single-point framework and CDR mutations from a high nanomolar binder (UCA_2-17).

(a) What fraction of the theoretical library was explored, and were these rationally or randomly chosen?

The fraction explored was 318 paired variants / 1,344 theoretical variants (23.7%). The subset we chose to explore was randomly chosen during library bottlenecking which we have now clarified in the main text in lines 325-326.

Then, in the next section, they mention "single and double mutants" of the 222-1C06 CDRH2.

(b) Were the 222-1C06/319-345 influenza bnAb libraries made using the same mature/UCA chimera strategy as the SARS-CoV-2 binders? If so, it may be clearer to present these two results in a different order or using a different linking phrase on page 7 line 298. Otherwise, if they were made using a different approach, please could the authors clarify what library construction strategy they used?

We agree that this section was confusing (Reviewer #1 also noted this; see response above).

3. Please could the authors comment on whether there are any limitations on the diversity of epitopes that can be studied in a single experiment? For example, if an antibody lineage's binding mode to an epitope sterically overlaps with another antibody lineage's mode to a nearby epitope, does this not risk distorting the fluorescence signal for the weaker binder (via competitive inhibition)?

This is a good technical point also raised by reviewer #1. As discussed in further detail in response to reviewer #1, we minimize multiple epitope binding across yeast cells by labeling cells with at least a 10-fold molar excess of each antigen per total displayed protein. We also gate explicitly on single cells using the gating strategy shown in Extended Data Figure 6.

The broader point, as we read it, relates to the total number and diversity of antigenic epitopes assessed by our current strategy. Theoretically, there is no upper limit. However, the practical limit on the number of antigens is likely three, for the points raised below.

Practical issues include: (i.) potential for antibody non-specific binding; (ii.) QC issues with ensuring multiple antigens are properly folded; and (iii.) changing antigen concentrations such that there is an appreciable binding signal for the populations at each condition tested. For an example of this latter point, in our implementation we mixed high HA labeling conditions with low S1 labeling conditions, and vice versa.

Minor:

Page 3, line 120: in this context, I suggest changing "epitopes" to "domains"; there are several classes of epitope within the RBD (10.1016/j.cell.2021.02.032).

We have made this change on line 130.

Page 4, line 184: biophysical -> biophysically

We have made this change on line 201.

Page 6, line 270: Should S74F be in this set?

The reviewer is correct, and the original text is incorrect. We have updated the text by removing S74F (line 291).

Page 8, line 346: “four variants” but only three mutations listed

We clarified the language here to show that there are four distinct variants (line 376).

REVIEWERS' COMMENTS

Reviewer #1 (Remarks to the Author):

I am fully satisfied with the authors edits and explanations to my critique (as well as the critique by reviewer #2). The manuscript is appropriate for publication in NC

Reviewer #2 (Remarks to the Author):

My comments on the previous manuscript have been fully addressed in the response and revised version. I have no further concerns or suggestions, and commend the authors on an excellent study.

Reviewer #2 (Remarks on code availability):

Reviewed in first round of review, no changes required.